# Spatial-Aware Decision-Making with Ring Attractors in Reinforcement Learning Systems

**Marcos N. Saura**
The University of Manchester
Manchester, United Kingdom
marcos.negresaura@postgrad.manchester.ac.uk

**Richard Allmendinger**
The University of Manchester
Manchester, United Kingdom
Richard.Allmendinger@manchester.ac.uk

**Wei Pan**
The University of Manchester
Manchester, United Kingdom
wei.pan@manchester.ac.uk

**Theodore Papamarkou**
PolyShape
Athens, Greece
theodore@polyshape.com

## Abstract

Ring attractors, mathematical models inspired by neural circuit dynamics, provide a biologically plausible mechanism to improve learning speed and accuracy in Reinforcement Learning (RL). Serving as specialized brain-inspired structures that encode spatial information and uncertainty, ring attractors explicitly encode the action space, facilitate the organization of neural activity, and enable the distribution of spatial representations across the neural network in the context of Deep Reinforcement Learning (DRL). These structures also provide temporal filtering that stabilizes action selection during exploration, for example, by preserving the continuity between rotation angles in robotic control or adjacency between tactical moves in game-like environments. The application of ring attractors in the action selection process involves mapping actions to specific locations on the ring and decoding the selected action based on neural activity. We investigate the application of ring attractors by both building an exogenous model and integrating them as part of DRL agents. Our approach significantly improves state-of-the-art performance on the Atari 100k benchmark, achieving a 53% increase in performance over selected baselines.

## 1 Introduction

This paper addresses the challenge of efficient action selection in Reinforcement Learning (RL), particularly in environments with spatial structures, ranging from robotic manipulation where joint movements are coupled, to game-playing agents where tactical decisions might depend on positional awareness. We integrate ring attractors, a neural circuit model from neuroscience originally proposed by [50] and later experimentally validated by [18], into the RL framework. This approach provides a mechanism for uncertainty-aware decision-making in RL, thereby yielding more efficient and reliable learning in complex environments. Ring attractors offer a unique framework to represent spatial information in a continuous and stable manner [33]. In a ring attractor network, neurons interconnect circularly, forming a loop with tuned connections [7]. This configuration allows for robust and localized activation patterns, maintaining accurate spatial representations even in the

39th Conference on Neural Information Processing Systems (NeurIPS 2025).

presence of noise or perturbations. Applying ring attractors to the selection of RL actions involves mapping actions to specific ring locations and decoding the selected action based on neural activity. This spatial embedding proves advantageous for continuous action spaces, particularly in tasks such as robotic control and navigation [28]. Ring attractors improve decision-making by exploiting spatial relations between actions, contributing to informed transitions between actions in sequential decision-making tasks in RL. While many action spaces possess inherent topological structure [39], traditional neural networks represent actions as orthogonal vectors that ignore these relationships. Ring attractors bridge this gap by providing a neural substrate that preserves spatial ordering and similarity between actions. This is particularly relevant given recent advances showing that spatial structure in representations improves sample efficiency [25, 49] and generalization [3, 13], yet these benefits have not been systematically applied to action selection mechanisms themselves.

In what follows, we summarize our contributions. Briefly, our contributions include a novel approach to RL policies based on ring attractors, the inclusion of uncertainty-aware capabilities in our RL systems, and the development of Deep Learning (DL) modules for RL with ring attractors.

- We propose a novel approach for integrating ring attractors into RL, incorporating spatial information, temporal filtering, and relations between actions. This spatial awareness significantly speeds up the learning rate of the RL agent. See Sections 3.1.2 and 4.1.

- We utilize a Bayesian approach to build ring attractors input signals, encoding uncertainty estimation to drive the action selection process, showing further performance improvement. See Sections 3.1.3 and 4.1.

- We develop a reusable DL module based on recurrent neural networks that incorporates ring attractors into DRL agents. This allows integration and comparison with existing DRL frameworks and baselines, enabling the adoption of our ring attractor approach across different RL models and tasks in applied domains. See Sections 3.2 and 4.2.

Codebase available at `https://github.com/marcosaura/RA_RL`.

## 2 Related Research

The integration of ring attractors into RL systems brings together neuroscience-inspired models and advanced machine learning techniques. In this section, we review the literature on key areas that form the foundation of our RL research: spatial awareness in RL, biologically inspired RL approaches, and uncertainty quantification methods.

### 2.1 Spatial Awareness in RL

Incorporating spatial awareness into RL systems has improved performance on tasks with inherent spatial structure. Regarding relational RL, [49] introduced an approach using attention mechanisms to reason about spatial relations between entities in an environment. This method demonstrated improved sample efficiency and generalization in tasks that require spatial reasoning. On the topic of navigation, [25] developed a DRL agent capable of navigating complex city environments using street-level imagery. Their approach incorporated auxiliary tasks, such as depth prediction and loop closure detection. Concerning explicit spatial representations, [13] proposed a cognitive mapping and planning approach for visual navigation, combining spatial memory with a differentiable neural planner. Similarly, [3] introduced a relational DRL framework using graph neural networks to capture spatial relations between objects. Although these approaches demonstrate the importance of spatial awareness in RL, they rely on emergent learning to develop spatial understanding through comprehensive architectural additions, which require extensive training to discover action relationships. In contrast, our ring attractor approach explicitly encodes spatial structure in the action space, providing spatial inductive bias that accelerates learning rather than relying on the RL agent to autonomously discover all critical action space relationships, which may result in suboptimal or incomplete spatial understanding.

### 2.2 Biologically Inspired Machine Intelligence

Biologically inspired approaches to RL seek to leverage insights from neuroscience to improve the efficiency, adaptability, and interpretability of RL algorithms. These methods often draw upon neural circuit dynamics and cognitive processes observed in biological systems. A notable example is

the work in [45], which demonstrated how neural attractor states implement probabilistic choices through recurrent network dynamics. The work presented in [2] demonstrated that incorporating grid-like representations, inspired by mammalian grid cells, into RL agents improved performance in navigation tasks. Their work showed that these biologically inspired representations emerged naturally in agents trained on robotics tasks and transferred well to new environments. Similarly, [10] showed that recurrent neural networks trained on navigation tasks naturally developed grid-like representations, suggesting a deep connection between biological and artificial navigation systems. In RL, grid cells and ring attractors serve distinct computational roles. Grid cells encode spatial state information (the agent's position), through hexagonal firing patterns, functioning as fixed basis functions for value functions and spatial generalization [2, 10]. Ring attractors encode persistent states, directional and spatial bias, through recurrent dynamics that actively maintain information over time [16]. While grid cells provide emergent spatial input features, ring attractors are dynamical components that provide intrinsic spatial encoding for memory states [17], that here we translate to encoding of the action space. [31] demonstrated how RL agents naturally develop insect-like behaviors and neural dynamics when solving complex spatial navigation tasks. [44] proposed a biologically inspired meta-RL algorithm that mimics the function of the prefrontal cortex and dopamine-based neuromodulation. Their approach demonstrated rapid learning and adaptation to new tasks, similar to the flexibility observed in biological learning systems. However, these approaches primarily focus on mimicking neural representations or learning dynamics, whereas ring attractors specifically encode spatial relationships in the action space itself, providing both biological plausibility and direct spatial awareness for action selection.

## 2.3   Uncertainty Quantification

Regarding exploration strategies, [27] introduced bootstrapped Deep Q-networks (DQNs), addressing exploration by leveraging uncertainty in Q-value estimates by training multiple DQNs with shared parameters. Building on this theme, [8] proposed random network distillation (RND), measuring uncertainty by comparing predictions between a target network and a randomly initialized network. For efficient uncertainty quantification, [11] and [9] presented a novel "masksemble" approach, applying masks across the input batch during the forward pass to generate diverse predictions. Addressing risk assessment in non-stationary environments, [14] described a method to analyze sources of lack of knowledge by adding a second Bayesian model to predict algorithmic action risks, particularly relevant for multi-agent RL (MARL) systems. [19] developed a Bayesian ring attractor that outperforms conventional ring attractors by dynamically adjusting its activity based on evidence quality and uncertainty. In the context of individual treatment effects, [20] performed uncertainty quantification (UQ) using an exogenously prescribed algorithm, making the method agnostic to the underlying recommender algorithm. [1] developed a Bayesian approach for RL in episodic high-dimensional Markov decision processes (MDPs). They introduced two novel algorithms: LINUCB and LINPSRL. These algorithms achieve significant improvements in sample efficiency and performance by incorporating uncertainty estimation into the learning process. In the context of DRL, Bayesian Deep Q-networks (BDQNs) [1] incorporate efficient Thompson sampling and Bayesian linear regression at the output layer to factor uncertainty estimation in the action-value estimates. While these methods treat uncertainty estimation as a separate module applied to existing architectures, our ring attractor approach integrates uncertainty through the variance parameters of Gaussian input signals, making uncertainty awareness an intrinsic part of the spatial action representation.

In summary, the literature reveals a growing interest in incorporating spatial awareness, biological inspiration, and UQ into RL systems. However, there remains a gap in integrating these elements into a cohesive framework that simultaneously provides biological plausibility, explicit spatial encoding, and natural uncertainty handling within a single neural architecture. Our work on ring attractors aims to bridge this gap by providing a biologically plausible model that inherently captures spatial relations and can be extended to handle uncertainty, potentially leading to more robust and efficient RL agents.

## 3   Methodology

Ring attractors represent a computational principle where neurons are arranged in a circular topology to maintain spatial representations. Originally proposed by [50] for encoding heading direction and empirically discovered by [18] in Drosophila, these networks exhibit circular organization where head direction cells encode different spatial directions [38]. For our methodology, ring attractors

integrate multiple cues through distance-weighted connections [51, 12] and maintain representational stability through excitatory-inhibitory dynamics [37]. This bioinspired structure has been repurposed in this line of research to explicitly encode the action space during the decision-making process. In this section, we describe two main methods: an exogenous ring attractor model using continuous-time recurrent neural networks (CTRNNs) and a DL-based ring attractor integrated into the RL agent. While the CTRNN model validates the theoretical principles, the DL architecture eases deployment and adoption in existing DRL frameworks. Both leverage the ring attractors' spatial encoding capabilities to enhance action selection and performance. In the CTRNN model, we detail the ring attractor architecture, Section 3.1.1; dynamics and implementation, Section 3.1.2; and uncertainty injection, Section 3.1.3. CTRNNs are used for their ability to model continuous neural dynamics and maintain stable attractor states [4]. The integrated DL approach offers end-to-end training for efficiency and scalability, detailed in Section 3.2. Ring attractors in RL maintain stable spatial information representations, preserving action relations lost in traditional flattened action spaces. This circular spatial representation potentially yields smoother policy gradients and more efficient learning in spatial tasks, attributed to the ring attractors' ability to maintain a stable representation of spatial information.

## 3.1 Continuous-Time RNN Ring Attractor Model

During the first stage of the research, the focus is on developing a self-contained exogenous ring attractor as a CTRNN. This will be integrated into the output of the value-based policy model to perform action selection.

### 3.1.1 Ring Attractor Architecture

Ring attractors commonly consist of a configuration of excitatory and inhibitory neurons arranged in a circular pattern. We can model the dynamics of the ring using the Touretzky ring attractor network [40]. In this model, each excitatory neuron establishes connections with all other excitatory neurons, and an inhibitory neuron is placed in the middle of the ring with equal weighted connections to all excitatory neurons.

**Excitatory neurons' input signal**: Let $x_n^i \in \mathbb{R}^s$ denote the input signal from source number $i$ to the excitatory neuron $n = 1, \ldots, N$. The total input to neuron $n$ is defined as the sum of all input signals $I$ for that particular neuron: $x_n = \sum_{i=1}^{I} x_n^i$, where $x_n \in \mathbb{R}$. To model input signals $x_n^i$ of varying strengths, these signals are commonly viewed as Gaussian functions $x_n^i : \mathbb{R}^s \to \mathbb{R}^s$. These functions allow us to represent the input to each neuron as a sum of weighted Gaussian distributions. The key parameters of these Gaussian functions are: $K_i$, the magnitude variable for the input signal in index $i$, which determines the overall strength of the signal; $\mu_i$, which defines the mean position of the Gaussian curve in the ring for the input signal $i$, representing the central focus of the signal; $\sigma_i$, the standard deviation of the Gaussian function, which determines the spread or reliability of the signal; and $\alpha_n$, which represents the preference for the orientation of the neuron $n$ in space. These parameters combine to the following:

$$x_n(\alpha_n) = \sum_{i=1}^{I} x_n^i(\alpha_n) = \sum_{i=1}^{I} \frac{K_i}{\sqrt{2\pi\sigma_i}} \exp\left(-\frac{(\alpha_n - \mu_i)^2}{2\sigma_i^2}\right) \tag{1}$$

**Neuron activation function**: We employ rectified linear unit (ReLU) function $f(x) = \max(0, x + h)$, where $h \in \mathbb{R}^+$ as activation function for each neuron, where $h$ is a threshold that introduces the non-linear behaviour in the ring.

**Excitatory neuron dynamics**: The dynamics of excitatory neurons in the ring is described as:

$$\frac{dv_n}{dt} \approx \frac{\Delta v_n}{\Delta t} = \frac{v_{n+\Delta t} - v_n}{\Delta t} = \frac{f(x_n + \epsilon_n + \eta_n)}{\tau} - v_n \tag{2}$$

In Eq. (2) $v_n \in \mathbb{R}$ represents the activation of the excitatory neuron $n$, $x_n$ is previously defined in Eq. (1) is the external input to the neuron $n$ of Eq. (1), $\epsilon_n \in \mathbb{R}$ represents the weighted influence of the other excitatory neurons activation, which is defined mathematically in Eq. (4). $\eta_n \in \mathbb{R}$ is the influence from the weighted inhibitory neuron activation to the target excitatory neuron $n$, and $\tau = \Delta t$ is the time integration constant. This equation captures the evolution of neuronal activation over time, considering both excitatory and inhibitory activations.

**Inhibitory neuron dynamics**. The activation of the inhibitory neuron, which regulates network dynamics, is described by:

$$\frac{\mathrm{d}u}{\mathrm{d}t} \approx \frac{\Delta u}{\Delta t} = \frac{u_{+\Delta t} - u}{\Delta t} = \frac{f(\epsilon_n + \eta_n)}{\tau} - u \qquad (3)$$

Here, $u \in \mathbb{R}$ represents the inhibitory neuron's activation output, $\epsilon_n$ is the weighted sum of excitatory activations where in this case $n$ is the inhibitory neuron, and $\eta_n \in \mathbb{R}$ is the weighted self-inhibition activation term. This equation models how the inhibitory neuron integrates inputs from the excitatory population and its own state.

**Synaptic weighted connections**: The influence between neurons decreases with distance, as modeled by the weighted connections. This weighted connection applies to both excitatory and inhibitory neurons. For excitatory neurons: $w^{(E_m \to E_n)} = e^{-d^2_{(m,n)}}$, where $d_{(m,n)} = |m - n|$ is the distance between neurons $m$ and $n$. For the inhibitory neuron: $w^{(I \to E_n)} = e^{-d^2_{(m,n)}} = e^{-1}$. Note that our model contains a single inhibitory neuron placed in the middle of the ring, with a distance of 1 unit to all excitatory neurons. The excitatory ($\epsilon_n$) and inhibitory ($\eta_n$) weighted connections are also known in the literature as neuron-proximal excitatory and inhibitory voltage or potential. These are then defined as follows:

$$\epsilon_n = \sum_{m=1}^{N} w^{(E_m \to E_n)}_{m,n} v_m \qquad \eta_n = w^{(I \to E_n)} u \qquad (4)$$

### 3.1.2 Ring Attractor as Behavior Policy in RL

To integrate the ring attractor model with RL, we need to establish a connection between the estimated value of state-action pairs and the input to the ring attractor network. This integration allows the ring attractor to serve as a behavior policy, guiding action selection based on the values learned.

**Excitatory neurons' action input signal:** We begin by reformulating the input function for a target excitatory neuron $n$ from Eq. (1). To integrate the ring attractor with RL, we reformulate the input signals by mapping $K_i$ to the Q-value $Q(s, a)$, effectively substituting input signal index $i \in I$ with actions $a \in A$. In other words, the key modification is setting the scale factor $K_i$ to the Q-value $Q(s, a)$ of the state-action pair $(s, a)$, that is $K_i = Q(s, a)$.

This formulation is represented in Fig. 1 and described in Eq. (5) ensures that actions with higher estimated values are given more weight in the ring attractor dynamics, naturally biasing the network towards more valuable actions. The orientation of the signal within the ring attractor is determined by the direction of movement in the action space. We represent this as $\mu_i = \alpha_a(a)$, where $\alpha_a(a)$ is the angle corresponding to the action $a$ in the circular action space. We define our circular action space $\mathcal{A}$ as a subset of $\mathbb{R}^2$, where each action $a \in \mathcal{A}$ is represented by a point on the unit circle. The function $\alpha : \mathcal{A} \to [0, 2\pi)$ maps each action to its corresponding angle on this circle, and $\alpha_n$ which presents the preference for the orientation of the neuron $n$ in space. To account for uncertainty in our value estimates, we incorporate the variance of the estimated value for each action $\sigma_i = \sigma_a$.

This allows the network to represent not just the expected value of actions, but also the confidence in those estimates. Combining these, we arrive at the following equation for the action signal $x_n(Q)$:

$$x_n(Q(s, a)) = \sum_{a=1}^{A} \frac{Q(s, a)}{\sqrt{2\pi\sigma_a}} \exp\left(-\frac{(\alpha_n - \alpha_a(a))^2}{2\sigma_a^2}\right) \qquad (5)$$

This equation represents the input to each neuron as a sum of Gaussian functions, where each function is centered on an action's direction and scaled by its Q-value.

**Full excitatory neuron dynamics**. The dynamics of the excitatory neurons in the ring attractor, now incorporating the Q-value inputs, are described by:

$$\frac{\mathrm{d}v_n}{\mathrm{d}t} = \frac{1}{\tau}\left(max\left(0, \left(\sum_{m=1}^{m=N} w^{(E \to E)}_{m,n} v_m + x_n(Q) + w^{(I \to E_u)} u\right)\right)\right) - v_n \qquad (6)$$

This equation captures how the activation of each neuron evolves over time, influenced by the Gaussian functions of the input action value $x_n(Q)$, the excitatory feedback $\epsilon_n = \sum_{m=1}^{N} w^{(E_m \to E_n)} v_m$, and the inhibitory feedback $\eta_n = w^{(I \to E_n)} u$.

**Full inhibitory neuron dynamics**. The complete dynamics of the inhibitory neuron are given by:

$$\frac{\mathrm{d}u}{\mathrm{d}t} = \frac{1}{\tau}\left(max\left(0, \left(u + \sum_{n=1}^{N} w^{(E_n \to I)}_n v_n\right)\right)\right) - u, \qquad (7)$$

where the self-inhibition term $u$ and excitatory input $v_n$ from each excitatory neuron $n$ from Eq. (6) updates the activation of the inhibitory neuron $\frac{du}{dt}$. Eq. (6) and Eq. (7) collectively describe the dynamics of the ring attractor network, capturing the interplay between excitatory and inhibitory neurons, external inputs, and synaptic connections. To translate the ring attractor's output into an action in the 2D space, we use the following equation:

$$\text{action} = \text{argmax}\{\mathbf{V}\}\frac{N^{(A)}}{N^{(E)}}, \qquad (8)$$

where $n \in \{1, ..., N^{(E)}\}$, $N^{(E)}$ is the number of excitatory neurons in the ring attractor, $N^{(A)}$ is the number of discrete actions in the action space $\mathcal{A}$, and the activation of excitatory neurons around the ring $\mathbf{V} = [v_1, v_2, ..., v_{N^{(E)}}]$. This equation assumes that both the neurons in the ring attractor and the actions in the action space are uniformly distributed. This approach allows for nuanced action selection that takes into account both the spatial relations between actions and their estimated values. A visualization of the ring is presented in Fig. 1.

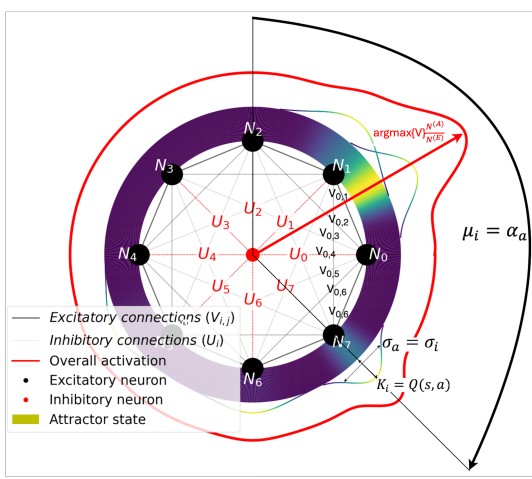

Figure 1: Ring attractor Touretzky representation: Circular arrangement of excitatory neurons (N0-N7) and central inhibitory neuron. Four input signals shown as colored gradients. Overall activation depicted by red outline. Includes connection weights and input signal parameters, illustrating the final ring attractor dynamics state from Eq. (8).

### 3.1.3 Uncertainty Quantification Integration

In the field of DRL, for any state-action pair $(s, a)$, the Q-value $Q(s, a)$ can be expressed as a function of the input state through a function approximation algorithm $\Phi_\theta(s)$ taking as input the current state $s$. This function approximation algorithm ($\Phi_\theta(s)$) can be expressed as the weight matrix of our function approximation algorithm transposed $\theta^T$ times the feature vector extracted from the input state $x(s)$: $Q(s, a) = \Phi_\theta(s) = \theta^T x(s)$ [34]. For further context on function approximation in RL see Appendix A.2. As stated in Section 3.1.2, the variance of the Gaussian functions, input to the ring attractor, will be given by the variance of the estimate value for that particular action $\sigma_i = \sigma_a$. Among the various methods to compute the uncertainty of the action ($\sigma_a$) we have chosen to compute a posterior distribution with Bayesian linear regression (BLR). BLR acts as output layer for our neural network (NN) of choice.

We choose a linear regression model because it does not compromise the efficiency of the NN, while at the same time it provides a distribution to compute the variance for the state-action pairs. The implementation is based on [1], where a Bayesian value-based DQN model was instantiated to output an uncertainty-aware prediction for the state-action pairs. The new $Q$ function is defined as:

$$Q(s, a) = \Phi_\theta(s)^T w_a, \qquad (9)$$

where $w_a$ are the weights from the posterior distribution of the BLR model. Eq. (9) represents the parameters of the final Bayesian linear layer. When provided with a state transition tuple $(s, a, r, s')$, where $s$ is the current state, $a$ is the action taken, $r$ is the reward received, and $s'$ is the next state. This tuple represents

**Algorithm 1** CTRNN Ring Attractor Action Selection
***
**Require: Input:** State $s$, action $a$ from Action space $\mathcal{A}$.
1: **Step 1:** Compute Q-values and Uncertainty
2: Compute mean $\bar{Q}(s, a)$ and variance $\sigma_a^2$ using Eq. 11
3: **Step 2:** Generate Input Action Signals
4: **for** each excitatory neuron $n$ **do**
5:      Generate Gaussian action signal using Eq. 5
6: **end for**
7: **Step 3:** Reach Attractor State
8: **for** timestep $t$ until $T$ **do**     ▷ *choose T=50 empirically*
9:      **for** each excitatory neuron $n$ **do**
10:         $\frac{dv_n}{dt}$ Update excitatory neurons using Eq. 6
11:      **end for**
12:      $\frac{du}{dt}$ Update inhibitory neuron using Eq. 7
13: **end for**
14: **Step 4:** Translate Neural Activity $V$ to *action* selected from action space $\mathcal{A}$ using Eq. 8
15: **return** action
***

a single step of interaction between agent and environment. The model learns to adjust the weights

$w_a$ of the BLR and the function approximation algorithm, i.e. neural networks (NNs) ($\Phi_\theta$), to align the Q values with the optimal action $a = argmax(\gamma\Phi_\theta(s)^T w_a)$,

$$Q(s,a) = \Phi_\theta(s)^T w_a \rightarrow y := r + \gamma\Phi_{\theta_{\text{target}}}(s')^T w_{\text{target}}\hat{a}, \tag{10}$$

where $\gamma$ is the discount factor, $y$ is the expected Q-value, $\Phi_{\theta_{\text{target}}}$ are the features from the next state $s'$ extracted by the function approximation algorithm $\Phi$ using the target network parameters, $\theta_{target}$ refers to the target function approximation algorithm parameters, and $\hat{a}$ is the predicted optimal action in the next state $s'$. The construction of the Gaussian BLR prior distribution and the weights $i$ sample $w_{a,i}$ collected from the posterior distribution are performed through Thompson Sampling. This process allows us to incorporate uncertainty into our action-value estimates. For details on the construction of the Gaussian prior distribution and the specifics of the sampling process, we refer the reader to [1]. Both the mean $\bar{Q}(s,a)$ and the variance $\bar{\sigma}_a^2$ of Eq. (11) are calculated from a finite number of samples $I$.

$$\bar{Q}(s,a) = \frac{\sum_{i=0}^{i=I} Q(s,a)_i}{I} = \frac{\sum_{i=0}^{i=I} w_{a,i}^T\Phi_\theta(s_t)}{I}, \bar{\sigma}_a^2 = \frac{\sum_{i=0}^{i=I}\left(w_{a,i}^T\Phi_\theta(s_t) - \mu_a\right)^2}{I-1} \tag{11}$$

## 3.2 Deep Learning Ring Attractor Model

To further enhance the ring attractor's integration into RL frameworks and agents, we provide a Deep Learning (DL) implementation. This approach improves model learning and integration with DRL agents. Our implementation offers both algorithmic improvements, by benefiting from DL training process, and software integration improvements, easing the deployment processes. Recurrent Neural Networks (RNNs) offer an approach for integrating ring attractors within DRL agents. Recent studies by [22] show that RNNs perform well in modeling sequential data and in capturing temporal dependencies for decision-making. Like CTRNNs, RNNs mirror ring attractors' temporal dynamics, with their recurrent connections and flexible architecture emulating their interconnected nature. Allowing modelling weighted connections for forward and recurrent hidden states, as shown in Appendix A.6. The premises for modeling RNN are as follows.

**Attractor state as recurrent connections**. RNN recurrent connections model the attractor state, integrating information from previous time steps into the current network state, allowing for the retention of information over time.

**Signal input as a forward pass**. Forward connections from previous layers are arranged circularly, mimicking the ring's spatial distribution. The attractor state encodes the task context, influenced by the current input and hidden state. A learnable time constant $\tau$, inherited from Eq. (2), controls input and temporal evolution, enabling adaptive behavior and elastic contribution to the attractor state.

### 3.2.1 DRL Agent Integration

To shape circular connectivity within a RNN, the weighted connections in the input signal $V(s)$ and the hidden state or attractor state $U(v)$ are computed as in (12). This circular structure mimics the arrangement of excitatory neurons in the ring attractor. Eq. (12) shows the input signal to the recurrent layer $V_{m,n}$ from neuron $m$ from the previous layer in the DL agent to neuron $n$ in the RNN. The hidden state, $U_{m,n}$ mimics an attractor state, representing the recurrent connections in the RNN. The weighted RNN connections include fixed input-to-hidden connections ($w^{I\rightarrow H}m,n$) to maintain the ring's spatial structure, and learnable hidden-to-hidden connections ($w^{H\rightarrow H}m,n$) to capture emerging action relationships. These depend on a parameter $\lambda$ that drives the decay over distance and distance between neurons $d(m,n)$ where $N$ is the total number of neurons for the RNN and $M$ is the count of neurons in the previous layer of the NN architecture. The function $\phi_\theta(s): \mathbb{R}^S \rightarrow \mathbb{R}^M$ maps the input state $s$ of the DL agent to a representation of characteristics that will be the input of the recurrent layer. Likewise, $\theta$

$$V(s)_{m,n} = \frac{1}{\tau}\Phi_\theta(s)^T w_{m,n}^{I\rightarrow H} = \frac{1}{\tau}\Phi_\theta(s)^T e^{-\frac{d(m,n)}{\lambda}}$$

$$d(m,n) = \min\left(|m - n\frac{M}{N}|, N - |m - n\frac{M}{N}|\right)$$

$$U(v)_{m,n} = h(v)^T w_{m,n}^{H\rightarrow H} = h(\Phi_\theta)^T e^{-\frac{d(m,n)}{\lambda}}$$

$$d(m,n) = \min\left(|m - n|, N - |m - n|\right) \tag{12}$$

represents the parameters of this function (i.e., the weights and biases of the NN layers preceding the RNN layer, which extract relevant features from the input). The function $h(v): \mathbb{R}^N \rightarrow \mathbb{R}^N$ is parameterized by learnable weights that map the information from previous forward passes to

the current hidden state. The learnable parameter $\tau$ is the positive time constant responsible for the integration of signals in the ring. Here, $\tau$ acts as a learnable scaling factor that controls the balance between immediate input responsiveness and temporal integration within the ring attractor dynamics, with smaller values increasing sensitivity to current inputs and enabling faster adaptation, while larger values promote stability and smoother transitions between attractor states by integrating information over longer time horizons. It defines the contribution of input $\phi_\theta(s)$ to the current hidden state, imitating the attractor state, applied to NNs.

Finally, the action-value function $Q(s,a)$ is derived from the RNN layer's output by applying the neurons activation function to the combined input $V(s)$ and hidden state information $U(s)$. The activation function of choice is a hyperbolic tangent $\tanh$, this function is symmetric around zero, leading to faster convergence and stability. However, the output range of $\tanh$ (-1 to 1) is not fully compatible with value-based methods, where the DL agents need to output action-value pairs in the range of the environment's reward function. To address this issue and prevent saturation of the $\tanh$ activation function, we scale the action-value pairs by multiplying them by a learnable scalar $\beta$, as $Q(s,a) = \beta h_t = \beta \tanh(V(s) + U(v)) = \beta \tanh\left(\frac{1}{\tau}\Phi_\theta(s_t)^T w^{I \to H} + h_{t-1}(v)^T w^{H \to H}\right)$. While our main focus is on value-based ring-attractor agents, we also provide an overview of methods and results for policy-gradient approaches in Appendix A.3.

## 4   Experiments

This section presents the findings of our experiments that validate our proposed approach to integrate ring attractors into RL algorithms. To assess the effectiveness of our method, we conducted comparisons between multiple baseline models and action spaces. The evaluation encompasses two implementations: a CTRNN exogenous ring attractor and a DL approach where the ring attractor is modeled directly into DRL agents. In both implementations, the action-value pairs $Q(s,a)$ are evenly distributed across the ring circumference. For the exogenous model, each action is associated with a specific discrete angle or continuous space, section 3.1.2. In the DL implementation, each neuron in the RNN corresponds to one action-value. The ring attractor serves as the output layer of the DL agent with the weights modeling the circular topology of the action space, Section 3.2.1. For both approaches, ring attractor agents are annotated with the suffix $RA$. We demonstrate that ring attractors significantly enhance action selection and speed up the learning process.

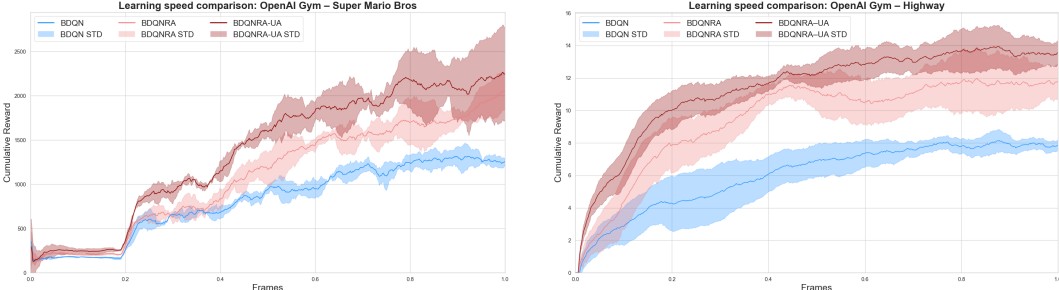

Figure 2: Learning speed comparison. Above: OpenAI Gym Super Mario Bros environment [15] with discrete action space. Below: OpenAI highway [21] with a continuous 1-D circular variable. The plot shows cumulative reward over 1 million frames for three models: Standard BDQN; BDQNRA with ring attractor behavior policy from Section 3.1.2, setting the action-value pair variance constant to $\sigma_a = \frac{\pi}{6}$, using this fixed variance to enable smooth action transitions while preventing interference with opposing actions; and BDQNRA-UA with RA and Uncertainty Awareness (UA) implementing the uncertainty quantification model from Section 3.1.3 to feed into the variance of the action-value pairs. Displaying mean episodic returns over 10 averaged seeds.

### 4.1   CTRNN Ring Attractor Model Performance

To evaluate our CTRNN ring attractor model from Section 3.1.2, integrated with BDQN [1] we performed experiments in the OpenAI Super Mario Bros environment [15] and OpenAI highway [21]. These benchmarks exhibit a spatially distributed complex decision-making scenario with 8 discrete different actions and a navigation-centric task with a continuous 1D circular actions space, respectively. Fig. 2 shows that both ring attractor models (BDQNRA and BDQNRA-UA) consistently outperform standard BDQN. The uncertainty-aware version (BDQNRA-UA) shows the best overall performance, highlighting the benefits of combining ring attractors spatial distribution of the action space with uncertainty-aware action selection. Empirical evaluations revealed that CTRNN-based ring attractor

models exhibited a mean computational overhead of $297.3\%$ (SD = $14.2\%$) compared to the baseline, significantly impacting runtime. To address this performance bottleneck and integrate the spatial understanding of the ring attractor into DRL, we developed a DL implementation of the ring attractor. This DL implementation is evaluated in the following subsections.

## 4.2 DL Ring Attractor Model Performance

To evaluate the effectiveness of our DL-based ring attractor implementation from Section 3.2, we conducted experiments implementing DDQNRA, an extension of DDQN [43]. Fig. 3 shows DDQNRA consistently outperforms standard DDQN, with the ring attractor's spatial encoding ability contributing significantly to improved learning speed. These results demonstrate substantial performance gains, especially in environments with strong spatial and directional cues.

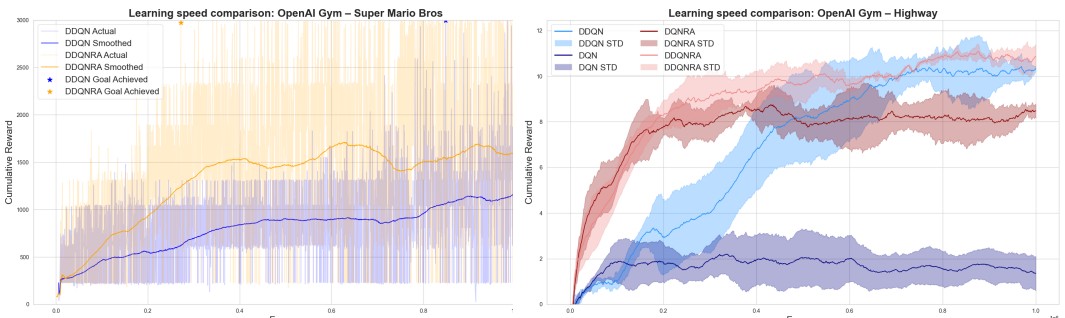

Figure 3: Performance comparison: DDQNRA vs standard DDQN [43] in two environments. Above: OpenAI Super Mario Bros [15], demonstrating adaptability to complex, game-like scenarios. Below: OpenAI highway [21], showing learning speed in spatial navigation tasks. Displaying mean episodic returns over 10 averaged seeds.

## 4.3 Performance on Atari 100k Benchmark

In this results section, we provide a comprehensive analysis of the performance of our DRL model on the Atari 100k benchmark [5]. We present comparisons with state-of-the-art models, highlighting the improvements achieved by our approach. Table 1 presents a comparison of our ring attractor-based DRL model integrated with Efficient Zero [48], which evaluates performance in Atari games with a limited training size of 100,000 environment steps. The table includes results from top-performing algorithms SPR [30] and CURL [32] for context. Our model demonstrates significant improvements over baseline methods, achieving a $53\%$ average improvement, with over $100\%$ gains in games with inherent spatial components, like Asterix and Boxing. Results demonstrate ring attractors provide consistent performance improvements across diverse RL tasks, with benefits extending beyond purely spatial tasks. We re-run baseline EffZero under identical cloud resources, to ensure fair comparison.

The mapping between game action spaces and ring configurations reflects the fundamental structure of each environment's action space. Games with primarily directional movement actions, such as *Asterix* and *Ms Pacman*, utilise a *Single* ring configuration where eight directional movements map naturally to positions around the ring circumference. In contrast, games combining movement with independent action dimensions, such as *Seaquest* and *BattleZone*, employ a *Double* ring configuration, one ring encoding movement actions and another representing secondary mechanics such as combat. This architecture maintains spatial relationships while preserving the independence of different action types. Further implementation details for multiple ring dynamics can be found in Appendix A.7.

Ablation studies were conducted to isolate the impact of key components in our ring attractor models, detailed in Appendix A.4.1 and A.4.2 for both implementations respectively. For the exogenous model, we compared performance with correct and randomized action distributions in the ring. In the DL implementation, we removed the circular weight distribution to assess its importance. Additionally, Appendix A.5 reveals preserved attractor dynamics behaving as a low-pass filter in the CTRNN and DL both at pre and post-training stages.

Table 1: Performance comparison on Atari 100k Benchmark [5]. Benchmark performed across all environments where actions can be laid out in one or more 2D action space planes (ring attractors). This is represented by the *ring* configuration column. Game score and overall mean and median human-normalized scores are recorded at the end of training and averaged over 10 random seeds, 3 samples per seed.

| Game Environment | Ring | Agent Human | CURL | Reported SPR | EffZero | Implemented EffZero | EffZeroRA |
|---|---|---|---|---|---|---|---|
| Alien | Double | 7127.7 | 558.2 | 801.5 | 808.5 | 738.1 | **1098.8** |
| Asterix | Single | 8503.3 | 734.5 | 977.8 | 25557.8 | 14839.3 | **31037.3** |
| Bank Heist | Double | 753.1 | 131.6 | 380.9 | 351.0 | 362.8 | **460.5** |
| BattleZone | Double | 37187.5 | 14870.0 | **16651.0** | 13871.2 | 11908.7 | 15672.0 |
| Boxing | Double | 12.1 | 1.2 | 35.8 | 52.7 | 30.5 | **62.4** |
| Chopper C. | Double | 7387.8 | 1058.5 | 974.8 | 1117.3 | 1162.4 | **1963.0** |
| Crazy Climber | Single | 35829.4 | 12146.5 | 42923.6 | 83940.2 | 83883.0 | **100649.7** |
| Freeway | Double | 29.6 | 26.7 | 24.4 | 21.8 | 22.7 | **31.3** |
| Frostbite | Double | 4334.7 | 1181.3 | 1821.5 | 296.3 | 287.5 | **354.8** |
| Gopher | Double | 2412.5 | 669.3 | 715.2 | 3260.3 | 2975.3 | **3804.0** |
| Hero | Double | 30826.4 | 6279.3 | 7019.2 | 9315.9 | 9966.4 | **11976.1** |
| Jamesbond | Double | 302.8 | 471.0 | 365.4 | **517.0** | 350.1 | 416.4 |
| Kangaroo | Double | 3035.0 | 872.5 | 3276.4 | 724.1 | 689.2 | **1368.8** |
| Krull | Double | 2665.5 | 4229.6 | 3688.9 | 5663.3 | 6128.3 | **9282.1** |
| Kung Fu M. | Double | 22736.3 | 14307.8 | 13192.7 | 30944.8 | 27445.6 | **49697.7** |
| Ms Pacman | Single | 6951.6 | 1465.5 | 1313.2 | 1281.2 | 1166.2 | **2028.0** |
| Private Eye | Double | 69571.3 | **218.4** | 124.0 | 96.7 | 94.3 | 155.8 |
| Road Runner | Double | 7845.0 | 5661.0 | 669.1 | 17751.3 | 19203.1 | **29389.3** |
| Seaquest | Double | 42054.7 | 384.5 | 583.1 | 1100.2 | 1154.7 | **1532.8** |
| **Human–normalised Score** | | | | | | | |
| | Mean | 1.000 | 0.428 | 0.638 | 1.101 | 0.959 | **1.454** |
| | Median | 1.000 | 0.242 | 0.434 | 0.420 | 0.403 | **0.531** |

## 5 Conclusion and Future Work

**Conclusion.** This paper presents a novel approach to RL, integrating ring attractors into action selection. Our work demonstrates that these neuroscience-inspired ring attractors significantly enhance learning capabilities for RL agents, leading to more stable and efficient action selection, particularly in spatially structured tasks.

The integration of ring attractors as a DL module proves to be particularly effective, allowing for end-to-end training and easy incorporation into existing RL architectures. Our experiments demonstrate significant improvements in action selection and learning speed, achieving state-of-the-art results on the Atari 100K benchmark. Notably, we observe an average 53% performance increase across all tested games compared to prior baselines. The most relevant improvements were observed in games with strong spatial components, such as Asterix (110 % improvement) and Boxing (105% improvement). Additionally, we observed improvements in other environments tested outside the Atari benchmark, further supporting the effectiveness of our approach across RL tasks and agents.

**Future work.** Future research should investigate scalability in high-dimensional action spaces and explore its efficacy in domains where spatial relationships are less straightforward. We believe that the success of this approach opens up several future research paths. This work can be extended to multi-agent scenarios and continuous control tasks as these have been the current limits of this piece of research. The ring attractor approach may exhibit reduced applicability in discrete action spaces with cardinality $|\mathcal{A}| < 3$ where the circular topology overhead exceeds potential spatial encoding benefits, and in action spaces where the action correlation matrix approaches the identity matrix, indicating absence of spatial dependencies or sequential relationships between discrete actions. Additionally, the extension to 3D spatial navigation tasks presents an organic expansion of this work, these scenarios will naturally introduce realistic sensor noise into the system.

In the field of uncertainty-aware decision making, leveraging the spatial structure provided by attractor networks presents a promising avenue to map uncertainty explicitly to the action space. Deploying the techniques presented here into specific domains could yield performance boosts, especially in safe RL, leveraging their stability properties to enforce constraints and ensure predictable behavior. This approach not only improves performance but also offers potential insight into spatial encoding of actions and decision-making processes, bridging the gap between neuroscience-inspired models and practical RL agents.

## Acknowledgments and Disclosure of Funding

This work was conducted as part of a self-funded PhD programme at the University of Manchester. Computational resources were provided by the Computational Shared Facility 3 (CSF3) at the University of Manchester [41]. We gratefully acknowledge the support of the Research IT team for maintaining and providing access to the GPU Cluster.

**Funding:** This research received no external funding.

**Competing Interests:** No competing interests are declared.

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

# A   Appendix

## A.1   Background: Attractor Networks Theoretical Foundations

Ring attractor networks are a type of biological neural structure that has been proposed to underlie the representation of various cognitive functions, including spatial navigation, working memory, and decision-making [18].

**Biological intuition**. In the early 1990s, the research carried out by [50] proposed that ring neural structures could underlie the representation of heading direction in rodents. [50] argued that the neural activity in an attractor network might encode the direction of the animal's head, with the network transitioning from one attractor state to another state as the animal turns.

**Empirical evidence**. There is growing evidence from neuroscience supporting the role of ring attractors in neural processing. For example, electrophysiological recordings from head direction cells (HDCs) of rodents have revealed a circular organization of these neurons, with neighbouring HDCs encoding slightly different heading directions [38]. Furthermore, studies have shown that HDC activity can be influenced by sensory inputs, such as visual signals and vestibular signals, and that these inputs can cause the network to update its representation of heading direction [37]. [47] showed model internal noise biases toward accuracy over speed, while environmental uncertainty exhibits a U-shaped effect, where moderate uncertainty favors speed and extreme uncertainty favors accuracy. [46] demonstrated that biological navigation networks rely on attractor dynamics while maintaining adaptability through continuous synaptic plasticity and sensory feedback.

**Sensor fusion in ring attractors**. Ring attractor networks provide a theoretical foundation for understanding cognitive functions such as spatial navigation, working memory, and decision-making. In the context of action selection in RL, sensor fusion plays a pivotal role in augmenting the information-processing capabilities of these networks. By combining data from various sensory modalities, ring attractors create a more nuanced and robust representation of the environment, essential for adaptive behaviors [24]. Research has elucidated the relationship between ring attractors and sensory inputs, with the circular organisation of HDCs in rodents complemented by the convergence of visual and vestibular inputs, highlighting the integrative nature of sensory information within the ring attractor framework [51].

**Modulation by sensory inputs**. Beyond the spatial domain, sensory input dynamically influences the activity of ring attractor networks. Studies have shown that visual cues and vestibular signals not only update the representation of heading direction but also contribute to the stability of attractor states, allowing robust spatial memory and navigation [12].

**Sensor fusion for action selection**. The concept of sensor fusion within the context of ring attractors extends beyond traditional sensory modalities, encompassing diverse sources, such as proprioceptive and contextual cues [23]. Building on the foundation of ring attractor networks discussed earlier, the integration of sensor fusion in the context of action selection involves fusing the action values associated with each potential action within the ring attractor framework. In particular, sensory information, previously shown to modulate the activity of ring attractor networks, extends its influence to the representation of action values. The inclusion of sensory information reflects a higher cognitive process, where the adaptable nature of ring attractor networks supports optimal decision-making and action selection in complex environments.

## A.2   Function Approximation for Q-Values

### A.2.1   Motivation and Background

In RL, an agent optimizes behaviour by interacting with an environment modeled as a Markov Decision Process (MDP), $\mathcal{M} = (\mathcal{S}, \mathcal{A}, P, R, \gamma)$, where $\mathcal{S}$ is the state space, $\mathcal{A}$ is the action space, $P : \mathcal{S} \times \mathcal{A} \times \mathcal{S} \to [0, 1]$ is the state transition probability function, $R : \mathcal{S} \times \mathcal{A} \times \mathcal{S} \to \mathbb{R}$ is the reward function, and $\gamma \in [0, 1)$ denotes the discount factor. The agent seeks to learn a policy $\pi(a \mid s)$, which defines the probability of selecting action $a$ in state $s$, that maximizes the expected discounted return [34]:

$$J(\pi) = \mathbb{E}_\tau \left[ \sum_{t=0}^{\infty} \gamma^t r_t \right], \tag{13}$$

where $J(\pi)$ is the objective function representing the expected total return (performance measure) of policy $\pi$, $\tau = (s_0, a_0, r_0, s_1, a_1, r_1, \ldots)$ represents a trajectory of states, actions, and rewards, $r_t$

is the reward received at time step $t$, $\gamma$ is the discount factor that balances immediate versus future rewards, and the expectation is taken over trajectories sampled by following policy $\pi$.

Action-value functions are foundational to various value-based RL algorithms. The action-value function is defined as:

$$Q^\pi(s, a) = \mathbb{E}_\tau \left[ \sum_{t=0}^{\infty} \gamma^t r_t \,\middle|\, s_0 = s,\, a_0 = a,\, \pi \right], \tag{14}$$

which quantifies the expected cumulative discounted reward when executing action $a$ in state $s$ and subsequently following policy $\pi$ for all future time steps.

This function satisfies a recursive relationship known as the Bellman equation [6]:

$$Q^\pi(s, a) = \mathbb{E}_{s', r \sim P, R} \left[ r + \gamma \sum_{a'} \pi(a' \mid s') Q^\pi(s', a') \right], \tag{15}$$

where $\mathbb{E}_{s', r \sim P, R}$ denotes expectation with respect to the next state $s'$ and reward $r$ following the environment dynamics defined by $P$ and $R$, and $r + \gamma \sum_{a'} \pi(a' \mid s') Q^\pi(s', a')$ is the expected immediate reward plus the discounted value of future actions. This forms the mathematical basis for value iteration, policy iteration, Q-learning, and other dynamic programming and temporal-difference methods used to compute or approximate $Q^\pi$.

In environments with large or continuous state-action spaces, storing tabular representations of $Q(s, a)$ becomes infeasible due to memory constraints and the inability to visit all state-action pairs sufficiently often. *Even in moderately sized discrete domains*, function approximation is often employed to leverage generalization across similar states or actions, thereby speeding up learning and reducing data requirements. By using function approximators, RL algorithms can handle high-dimensional inputs and share learned structure across different parts of the state-action space.

### A.2.2 General Formulation

Q-functions are approximated using parameterized functions $Q(s, a; \theta)$, where $\theta \in \mathbb{R}^d$ are learnable weights:

$$Q(s, a; \theta) \approx Q^\pi(s, a), \tag{16}$$

where the target $Q^\pi$ could be the Q-function of the current policy (policy evaluation) or the optimal Q-function $Q^*$ (control). The goal is to learn $\theta$ such that $Q(s, a; \theta)$ closely estimates the target Q-values across the relevant state-action space.

Parameters $\theta$ are typically updated through stochastic gradient descent to minimize a loss function. For instance, using the mean squared Bellman error:

$$\mathcal{L}(\theta) = \mathbb{E}_{s, a, r, s'} \left[ \left( r + \gamma \max_{a'} Q(s', a'; \theta') - Q(s, a; \theta) \right)^2 \right], \tag{17}$$

where $\theta'$ represents parameters of a target network that is periodically updated to match $\theta$ (in deep Q-learning algorithms such as DQN [26], or $\theta' = \theta$ in simpler settings like traditional Q-learning). The term $r + \gamma \max_{a'} Q(s', a'; \theta')$ is often called the "target" value, while $Q(s, a; \theta)$ is the "prediction". The parameters are updated according to:

$$\theta_{t+1} = \theta_t - \alpha \nabla_\theta \mathcal{L}(\theta_t), \tag{18}$$

where $\alpha$ is the learning rate, and $\nabla_\theta \mathcal{L}(\theta_t)$ is the gradient of the loss function with respect to $\theta_t$. In practice, these updates are performed using stochastic gradient descent or variants on mini-batches of experience to improve computational efficiency and stability.

### A.2.3 Linear Function Approximation

A straightforward approach is linear function approximation, where state-action pairs are mapped into a feature space via a feature function $\phi : \mathcal{S} \times \mathcal{A} \to \mathbb{R}^d$, and Q-values are computed as:

$$Q(s, a; \theta) = \theta^\top \phi(s, a). \tag{19}$$

This formulation is computationally efficient and, under specific conditions, enjoys theoretical convergence guarantees in on-policy settings [35]. However, linear approximation's representational

capacity is fundamentally constrained by the chosen feature representation $\phi(s, a)$. For discrete action spaces, an alternative formulation represents a separate linear function for each action:

$$Q(s, a; \theta) = \theta_a^\top \phi(s), \tag{20}$$

where $\phi(s) \in \mathbb{R}^d$ is a state feature vector and $\theta_a \in \mathbb{R}^d$ is a parameter vector specific to action $a$.

The transition from linear to neural approximation typically retains this foundational structure while enhancing the feature extraction process. Instead of relying on handcrafted features, neural networks learn feature representations automatically through their hidden layers.

### A.2.4 Neural Function Approximators

To capture richer, nonlinear dependencies, neural networks can be used to parameterize Q-functions. In the field of DRL, for any state-action pair $(s, a)$, the Q-value $Q(s, a)$ can be expressed as a function of the input state through a function approximation algorithm $\Phi_\theta(s)$. This notation evolves from the linear case but generalizes to allow for arbitrary nonlinear transformations of the input. Formally:

$$Q(s, a) = \Phi_\theta(s) = \theta^\top x(s), \tag{21}$$

where $x(s)$ corresponds to the (learned) feature representation of the state $s$ (e.g., the outputs of the penultimate layer), and $\theta$ represents the weights of the final layer.

With this design, neural networks learn both the feature extraction ($x(s)$) and the final mapping ($\theta$) end-to-end through backpropagation, enabling highly expressive function approximators that can generalize across large or complex state-action domains.

## A.3 Policy Gradient Methods with Ring Attractors

While previous sections focused on value-based methods, ring attractors can also enhance policy gradient methods, such as Proximal Policy Optimization (PPO) [29]. The integration follows similar principles, but adapts to the unique characteristics of policy-based approaches [36].

### A.3.1 Proximal Policy Optimization with Ring Attractors

In standard PPO, the policy is directly parameterized by a neural network that outputs action probabilities. When integrating ring attractors, we modify this structure to incorporate the spatial relationships between actions through the ring topology.

**Ring Attractor Policy Network.** For PPO integration, we adapt the DL ring attractor structure from Section 3.2. While the value-based methods use the network to process Q-values, in PPO we process policy logits. For a policy network with parameters $\theta$, the initial policy logits are computed as:

$$Z(s) = \Phi_\theta(s) \tag{22}$$

This parallels Section 3.2, where $Q(s, a) = \Phi_\theta(s)$ represents the output of the network for value-based methods. The difference is that $Z(s)$ represents policy logits rather than action-values.

These logits are then processed through the ring attractor using the same weighted connection structure. The input signal $V(s)$ uses $Z(s)$ as input and follows the same form as in Eq. (12):

$$V(s)_{m,n} = \frac{1}{\tau} Z(s)^T w_{m,n}^{I \rightarrow H} = \frac{1}{\tau} Z(s)^T e^{-\frac{d(m,n)}{\lambda}} \tag{23}$$

Similarly, the hidden state $U(v)$ maintains the same recurrent dynamics as in Eq. (12). The key difference occurs in the final output layer, where instead of scaling the output to represent Q-values, we apply a softmax function to directly obtain action probabilities:

$$\pi(a|s) = \text{softmax}(\tanh((V(s) + U(v)))) \tag{24}$$

### A.3.2 Policy Gradient Experimental Results

As shown in Figure 4, PPO [29] with ring attractors (PPO-RA) demonstrates significant performance improvements over standard PPO in the OpenAI Gym Super Mario Bros environment [15].

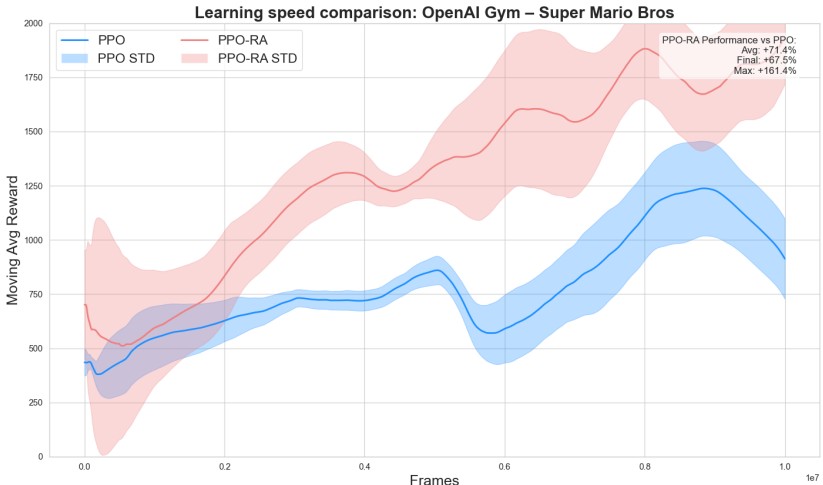

Figure 4: Performance comparison: PPO-RA vs standard PPO in OpenAI Gym Super Mario Bros environment [15] with discrete action space, demonstrating adaptability to complex, game-like scenarios. The plot shows PPO-RA (red) consistently outperforming standard PPO (blue), with shaded regions representing standard deviation across 10 averaged seeds.

Our experimental results reveal an average performance increase of $+71.4\%$ over standard PPO, with a final performance improvement of $+67.5\%$ and a maximum performance gain of $+161.4\%$ during training. These results demonstrate that the spatial awareness provided by ring attractors is beneficial across different RL paradigms, not just value-based methods. The performance gains in PPO-RA can be attributed to more efficient policy updates due to the preservation of spatial relationships between actions, smoother policy gradients resulting from the structured representation of the action space, and enhanced exploration through the natural diffusion of policy preferences across similar actions in the ring.

PPO-RA converges faster and achieves higher rewards than standard PPO. This suggests that ring attractors provide a general enhancement mechanism applicable to multiple RL approaches, showcasing that their utility may extend beyond the value-based methods explored in previous sections.

### A.4 Validating Ring Attractor Contributions Through Ablation Studies

### A.4.1 Exogenous Ring Attractor Model Ablation Study

To isolate the impact of the ring attractor structure, we conducted an ablation study comparing our full BDQNRA model against versions with the action space overlay in an incorrect distribution in the ring, Fig. 5. This incorrect distribution involves randomly rearranging the placement of actions within the ring, disrupting the natural topology of the action space.

For instance, this could mean placing opposing or unrelated actions side by side in the ring, such as pairing "move left" with "move down" instead of its natural opposite "move right". More generally, this incorrect distribution breaks the inherent relationships between actions that are typically preserved in the ring structure.

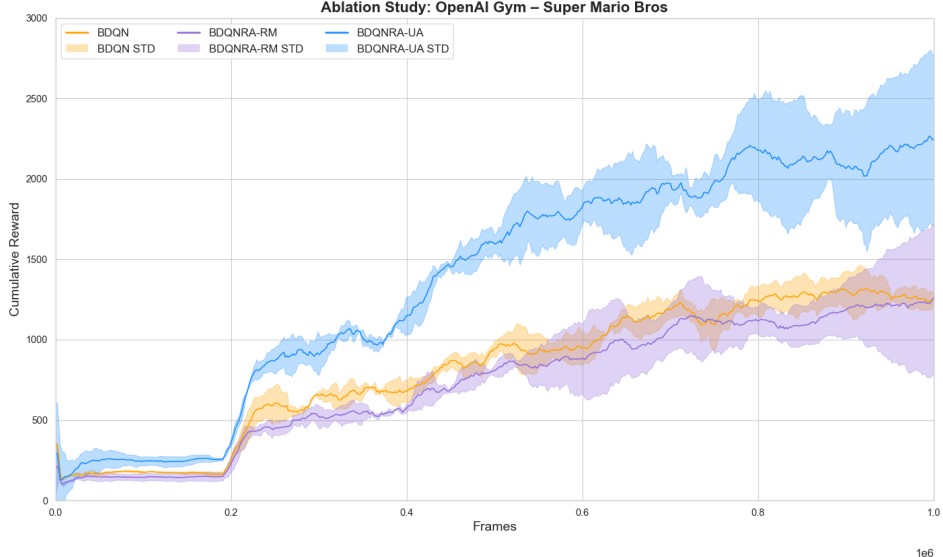

Figure 5: Ablation study comparing BDQN variants in OpenAI Gym Super Mario Bros [15]. The plot shows cumulative reward over 1 million frames for three models: Standard BDQN [1]; BDQNRA-UA with RA and Uncertainty Awareness (UA) implementing both the ring attractor behavior policy from Section 3.1.2 and the uncertainty quantification model from Section 3.1.3; and BDQNRA-RM, applying the same concepts from BDQNRA-UA, but randomly distributing the action space across the ring in each experiment. Displaying mean episodic returns over 10 averaged seeds.

### A.4.2 Deep Learning Ring Attractor Model Ablation Study

This ablation study focused on isolating the impact of the ring-shaped connectivity in our RNN-based ring attractor model. The key aspect of our experiment was to remove the circular weight distribution in both the forward pass (input-to-hidden connections) and the recurrent connections (hidden-to-hidden), while maintaining all other aspects of the RNN architecture. This approach allows us to directly assess the contribution of the spatial ring structure to the model's performance.

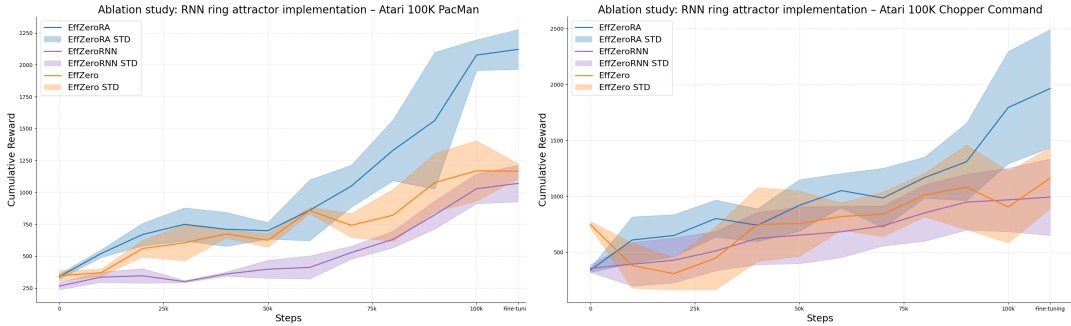

Figure 6: Ablation study results comparing the performance of the full RNN-based ring attractor model against a version with the circular weight distribution removed. The graph illustrates a significant performance drop for the Ms Pacman and Chopper Command environments in the Atari 100K benchmark [5]. This emphasizes the role of the circular topology in encoding spatial information and enhancing learning. Displaying mean episodic returns over 10 averaged seeds.

In our original model, the weights between neurons were determined by a distance-dependent function that created a circular topology. This function assigned stronger connections between neurons that were close together in the ring and weaker connections between distant neurons. For the ablation, we replaced this distance-dependent weight function with standard weight matrices for both the input-to-hidden and hidden-to-hidden connections. This modification effectively transforms our ring

attractor RNN into a standard RNN, where the weights are not constrained by the circular topology. We retained other key elements of the model, such as the learnable time constant and the non-linear transformation, to isolate the effect of the ring structure specifically.

### A.4.3 Deep Learning Ring Attractor Model Evolution

In this appendix section, we analyze model dynamics with both forward pass $(V(s))$ and hidden-to-hidden $(U(v))$ weights made trainable, rather than the standard approach of fixed forward pass connections, as presented in Section 3.2. As shown in Fig. 7, the forward-pass connections preserve the ring structure over training time, with strong distance-dependent decay patterns maintained throughout the learning process. This may indicate that the network naturally favors maintaining spatial topology for transmitting sensory information on a per-frame basis.

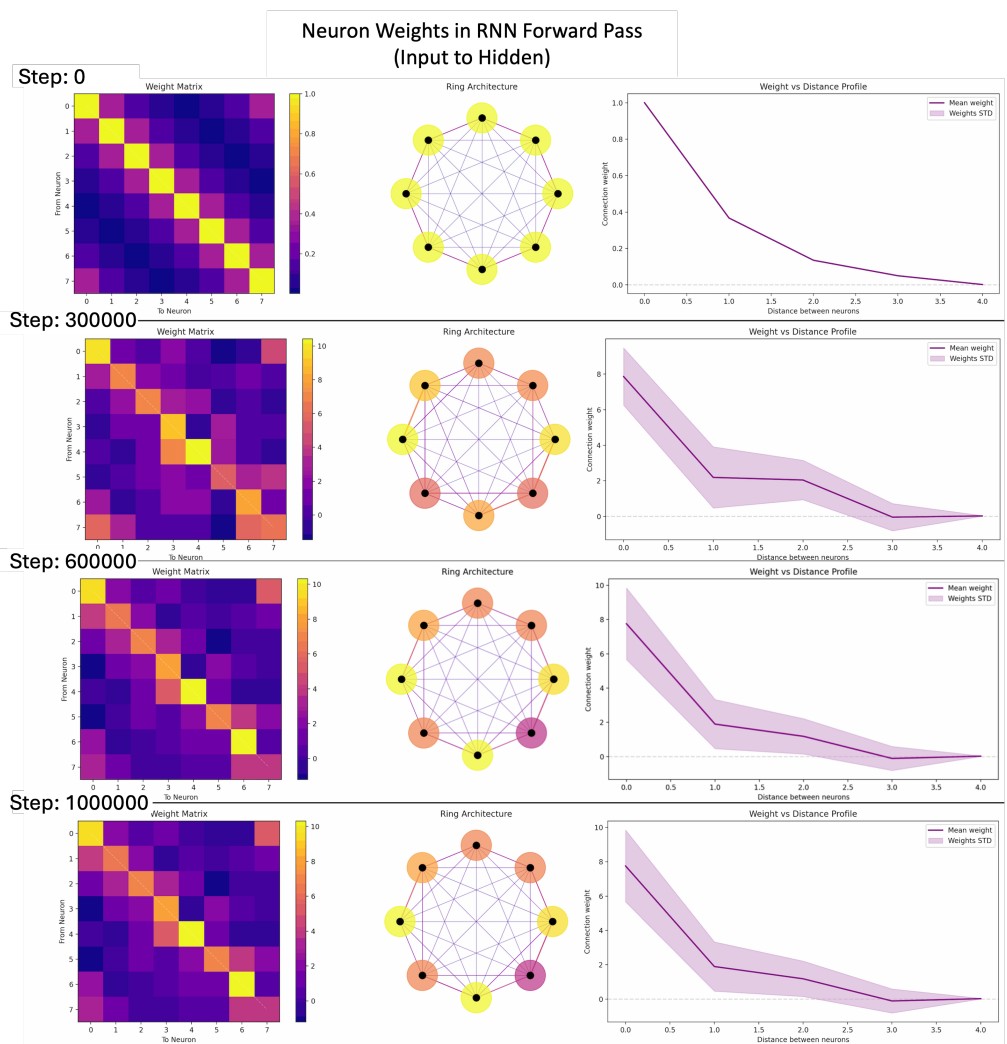

Figure 7: Evolution of forward pass weights showing preserved distance-dependent decay over training time, maintaining ring structure for spatial information transmission.

The hidden-to-hidden connections, depicted in Fig. 8, exhibit markedly different behavior. These connections evolve beyond their initial ring structure, developing specialized patterns that enable the encoding of environment-specific relationships between neurons in the hidden space. This flexibility in hidden-layer connectivity supports the learning of complex action relationships while building upon the structured spatial representation from the forward pass.

These findings validate our standard implementation approach described in Section 3.2, where forward pass connections are fixed and only hidden-to-hidden weights remain trainable. The natural preservation of ring structure in trainable forward weights suggests that this topology is inherently beneficial for processing spatial information, while adaptable hidden weights enable the task-specific learning demonstrated in our experimental results, Section 4.2.

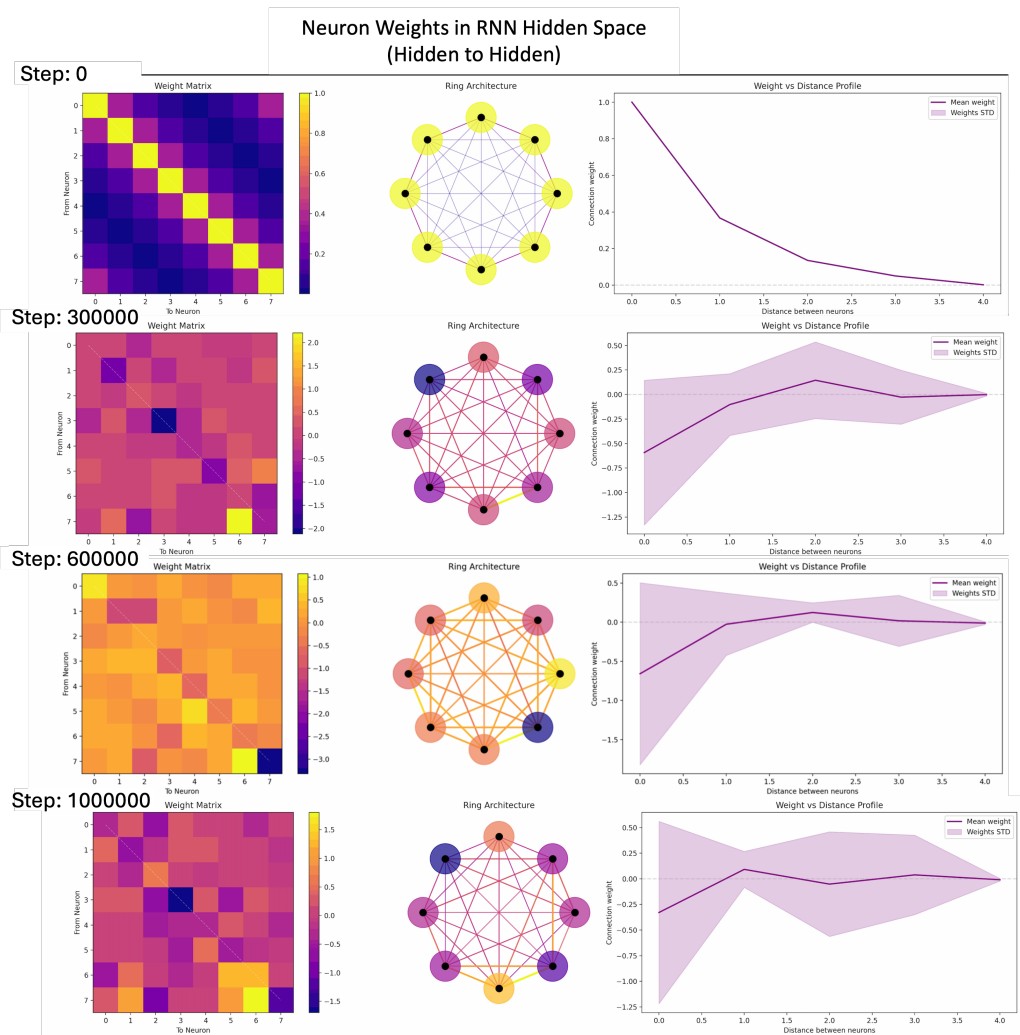

Figure 8: Development of hidden-to-hidden connections over time, demonstrating emergence of learned relationships between neurons beyond initial ring topology.

## A.5 Attractor Dynamics Validation

To validate the preservation of CANN dynamics in our DL implementation, we conducted controlled experiments examining the temporal evolution of neural states under dynamic input conditions. Ring attractors, as demonstrated in biological systems [18, 38], maintain stable activity patterns through recurrent connectivity that enables persistent neural states [16].

We present four experimental tables tracking hidden states and outputs across all eight neurons (DLN1-DLN8) in the ring attractor. Tables 2 and 3 examine pre-training dynamics under a two-stimulus protocol with asymmetric ($K_2 = 0.1 \cdot K_1$) and matched ($K_1 = K_2$) input amplitudes, respectively, demonstrating the network's ability to maintain stable attractor states and transition smoothly between them. Tables 4 and 5 present post-training dynamics after 315,000 and 750,000 training steps in the OpenAI highway [21], revealing how task-specific learning modifies the ring attractor structure while preserving attractor properties.

Table 2: Temporal dynamics of DL-based ring attractor with asymmetric input cues ($K_2 = 0.1 \cdot K_1$). The network maintains stable bumps and executes smooth transitions despite order-of-magnitude differences in input amplitude between the first cue (centered at neuron 2, steps 1-6) and second cue (centered at neuron 5, steps 7-14). Bold values indicate peak activation at each timestep.

| Step | Cue Ctr | CTRNN ArgMax | DL ArgMax | DLN1 In | DLN1 Hid | DLN1 Out | DLN2 In | DLN2 Hid | DLN2 Out | DLN3 In | DLN3 Hid | DLN3 Out | DLN4 In | DLN4 Hid | DLN4 Out | DLN5 In | DLN5 Hid | DLN5 Out | DLN6 In | DLN6 Hid | DLN6 Out | DLN7 In | DLN7 Hid | DLN7 Out | DLN8 In | DLN8 Hid | DLN8 Out |
|---|---|---|---|---|---|---|---|---|---|---|---|---|---|---|---|---|---|---|---|---|---|---|---|---|---|---|---|
| 1 | 2 | 2 | 2 | 1.63 | 0.75 | 52.28 | **1.73** | **0.78** | **55.21** | 1.63 | 0.75 | 52.28 | 1.38 | 0.67 | 44.98 | 1.09 | 0.58 | 36.86 | 0.93 | 0.55 | 32.47 | 1.09 | 0.58 | 36.86 | 1.38 | 0.67 | 44.98 |
| 2 | 2 | 2 | 2 | 1.63 | 0.82 | 53.83 | **1.73** | **0.86** | **56.97** | 1.63 | 0.82 | 53.83 | 1.38 | 0.72 | 46.02 | 1.09 | 0.61 | 37.38 | 0.93 | 0.56 | 32.75 | 1.09 | 0.61 | 37.38 | 1.38 | 0.72 | 46.02 |
| 3 | 2 | 2 | 2 | 1.63 | 0.71 | 51.50 | **1.73** | **0.75** | **54.70** | 1.63 | 0.71 | 51.50 | 1.38 | 0.60 | 43.54 | 1.09 | 0.49 | 34.74 | 0.93 | 0.44 | 30.05 | 1.09 | 0.49 | 34.74 | 1.38 | 0.60 | 43.54 |
| 4 | 2 | 2 | 2 | 1.63 | 0.65 | 50.12 | **1.73** | **0.69** | **53.34** | 1.63 | 0.65 | 50.12 | 1.38 | 0.54 | 42.11 | 1.09 | 0.42 | 33.27 | 0.93 | 0.37 | 28.56 | 1.09 | 0.42 | 33.27 | 1.38 | 0.54 | 42.11 |
| 5 | 2 | 2 | 2 | 1.63 | 0.64 | 49.94 | **1.73** | **0.68** | **53.17** | 1.63 | 0.64 | 49.94 | 1.38 | 0.53 | 41.92 | 1.09 | 0.41 | 33.06 | 0.93 | 0.36 | 28.35 | 1.09 | 0.41 | 33.06 | 1.38 | 0.53 | 41.92 |
| 6 | 2 | 2 | 2 | 1.63 | 0.65 | 50.14 | **1.73** | **0.69** | **53.37** | 1.63 | 0.65 | 50.14 | 1.38 | 0.54 | 42.12 | 1.09 | 0.42 | 33.26 | 0.93 | 0.37 | 28.55 | 1.09 | 0.42 | 33.26 | 1.38 | 0.54 | 42.12 |
| 7 | 5 | 2 | 2 | 0.14 | 0.65 | 17.51 | **0.15** | **0.70** | **18.60** | 0.15 | 0.65 | 17.75 | 0.16 | 0.54 | 15.42 | 0.16 | 0.43 | 12.89 | 0.16 | 0.37 | 11.68 | 0.15 | 0.43 | 12.74 | 0.15 | 0.54 | 15.17 |
| 8 | 5 | 2 | 2 | 0.14 | -0.02 | 2.72 | **0.15** | **-0.00** | **3.13** | 0.15 | -0.02 | 3.02 | 0.16 | -0.05 | 2.43 | 0.16 | -0.08 | 1.75 | 0.16 | -0.10 | 1.37 | 0.15 | -0.08 | 1.55 | 0.15 | -0.05 | 2.11 |
| 9 | 5 | 3 | 3 | 0.14 | -0.09 | 1.24 | 0.15 | -0.08 | 1.46 | **0.15** | **-0.08** | **1.56** | 0.16 | -0.09 | 1.50 | 0.16 | -0.10 | 1.34 | 0.16 | -0.10 | 1.19 | 0.15 | -0.10 | 1.13 | 0.15 | -0.09 | 1.15 |
| 10 | 5 | 3 | 4 | 0.14 | 0.01 | 3.28 | 0.15 | 0.01 | 3.43 | 0.15 | 0.01 | 3.60 | **0.16** | **0.01** | **3.70** | 0.16 | 0.01 | 3.61 | 0.16 | 0.01 | 3.61 | 0.15 | 0.01 | 3.48 | 0.15 | 0.01 | 3.34 |
| 11 | 5 | 4 | 5 | 0.14 | 0.06 | 4.50 | 0.15 | 0.07 | 4.64 | 0.15 | 0.07 | 4.82 | 0.16 | 0.07 | 4.97 | **0.16** | **0.07** | **5.01** | 0.16 | 0.07 | 4.94 | 0.15 | 0.07 | 4.79 | 0.15 | 0.06 | 4.61 |
| 12 | 5 | 5 | 5 | 0.14 | 0.07 | 4.66 | 0.15 | 0.07 | 4.79 | 0.15 | 0.07 | 4.98 | 0.16 | 0.08 | 5.14 | **0.16** | **0.08** | **5.19** | 0.16 | 0.08 | 5.13 | 0.15 | 0.07 | 4.97 | 0.15 | 0.07 | 4.78 |
| 13 | 5 | 5 | 5 | 0.14 | 0.06 | 4.48 | 0.15 | 0.06 | 4.61 | 0.15 | 0.07 | 4.80 | 0.16 | 0.07 | 4.96 | **0.16** | **0.07** | **5.02** | 0.16 | 0.07 | 4.96 | 0.15 | 0.07 | 4.80 | 0.15 | 0.06 | 4.60 |
| 14 | 5 | 5 | 5 | 0.14 | 0.06 | 4.36 | 0.15 | 0.06 | 4.48 | 0.15 | 0.06 | 4.68 | 0.16 | 0.06 | 4.84 | **0.16** | **0.06** | **4.90** | 0.16 | 0.06 | 4.84 | 0.15 | 0.06 | 4.68 | 0.15 | 0.06 | 4.48 |

Table 3: Temporal ring attractor dynamics with uniform input cues ($K_1 = K_2$). Bold values indicate peak activation at each timestep.

| Step | Cue Ctr | CTRNN ArgMax | DL ArgMax | DLN1 In | DLN1 Hid | DLN1 Out | DLN2 In | DLN2 Hid | DLN2 Out | DLN3 In | DLN3 Hid | DLN3 Out | DLN4 In | DLN4 Hid | DLN4 Out | DLN5 In | DLN5 Hid | DLN5 Out | DLN6 In | DLN6 Hid | DLN6 Out | DLN7 In | DLN7 Hid | DLN7 Out | DLN8 In | DLN8 Hid | DLN8 Out |
|---|---|---|---|---|---|---|---|---|---|---|---|---|---|---|---|---|---|---|---|---|---|---|---|---|---|---|---|
| 1 | 2 | 2 | 2 | 1.57 | 0.76 | 51.29 | **1.59** | **0.77** | **51.84** | 1.57 | 0.76 | 51.29 | 1.52 | 0.74 | 49.85 | 1.46 | 0.72 | 48.04 | 1.42 | 0.72 | 46.85 | 1.46 | 0.72 | 48.04 | 1.52 | 0.74 | 49.85 |
| 2 | 2 | 2 | 2 | 1.57 | 0.82 | 52.55 | **1.59** | **0.82** | **53.15** | 1.57 | 0.82 | 52.55 | 1.52 | 0.80 | 51.00 | 1.46 | 0.77 | 49.08 | 1.42 | 0.76 | 47.84 | 1.46 | 0.77 | 49.08 | 1.52 | 0.80 | 51.00 |
| 3 | 2 | 2 | 2 | 1.57 | 0.69 | 49.82 | **1.59** | **0.70** | **50.43** | 1.57 | 0.69 | 49.82 | 1.52 | 0.67 | 48.23 | 1.46 | 0.64 | 46.28 | 1.42 | 0.63 | 45.02 | 1.46 | 0.64 | 46.28 | 1.52 | 0.67 | 48.23 |
| 4 | 2 | 2 | 2 | 1.57 | 0.62 | 48.23 | **1.59** | **0.63** | **48.85** | 1.57 | 0.62 | 48.23 | 1.52 | 0.60 | 46.64 | 1.46 | 0.57 | 44.68 | 1.42 | 0.56 | 43.42 | 1.46 | 0.57 | 44.68 | 1.52 | 0.60 | 46.64 |
| 5 | 2 | 2 | 2 | 1.57 | 0.61 | 48.03 | **1.59** | **0.62** | **48.64** | 1.57 | 0.61 | 48.03 | 1.52 | 0.59 | 46.43 | 1.46 | 0.56 | 44.46 | 1.42 | 0.55 | 43.20 | 1.46 | 0.56 | 44.46 | 1.52 | 0.59 | 46.43 |
| 6 | 2 | 2 | 2 | 1.57 | 0.62 | 48.25 | **1.59** | **0.63** | **48.86** | 1.57 | 0.62 | 48.25 | 1.52 | 0.60 | 46.65 | 1.46 | 0.57 | 44.68 | 1.42 | 0.56 | 43.42 | 1.46 | 0.57 | 44.68 | 1.52 | 0.60 | 46.65 |
| 7 | 5 | 2 | 3 | 1.42 | 0.63 | 44.93 | 1.46 | 0.64 | 46.11 | **1.52** | **0.63** | **47.31** | 1.57 | 0.60 | 47.09 | 1.59 | 0.58 | 46.74 | 1.57 | 0.57 | 46.05 | 1.52 | 0.58 | 45.24 | 1.46 | 0.60 | 45.40 |
| 8 | 5 | 2 | 4 | 1.42 | 0.59 | 43.99 | 1.46 | 0.60 | 45.24 | 1.52 | 0.61 | 46.99 | **1.57** | **0.62** | **48.28** | 1.59 | 0.60 | 47.82 | 1.57 | 0.61 | 48.04 | 1.52 | 0.60 | 46.67 | 1.46 | 0.59 | 45.02 |
| 9 | 5 | 3 | 5 | 1.42 | 0.57 | 43.70 | 1.46 | 0.58 | 44.96 | 1.52 | 0.61 | 46.86 | 1.57 | 0.63 | 48.37 | **1.59** | **0.63** | **48.72** | 1.57 | 0.62 | 48.30 | 1.52 | 0.60 | 46.77 | 1.46 | 0.58 | 44.89 |
| 10 | 5 | 3 | 5 | 1.42 | 0.57 | 43.61 | 1.46 | 0.58 | 44.86 | 1.52 | 0.61 | 46.81 | 1.57 | 0.63 | 48.38 | **1.59** | **0.64** | **48.98** | 1.57 | 0.63 | 48.37 | 1.52 | 0.60 | 46.79 | 1.46 | 0.58 | 44.84 |
| 11 | 5 | 4 | 5 | 1.42 | 0.57 | 43.58 | 1.46 | 0.58 | 44.84 | 1.52 | 0.60 | 46.80 | 1.57 | 0.63 | 48.39 | **1.59** | **0.64** | **49.00** | 1.57 | 0.63 | 48.39 | 1.52 | 0.60 | 46.79 | 1.46 | 0.58 | 44.83 |
| 12 | 5 | 5 | 5 | 1.42 | 0.57 | 43.57 | 1.46 | 0.58 | 44.83 | 1.52 | 0.60 | 46.80 | 1.57 | 0.63 | 48.39 | **1.59** | **0.64** | **49.01** | 1.57 | 0.63 | 48.39 | 1.52 | 0.60 | 46.80 | 1.46 | 0.58 | 44.83 |
| 13 | 5 | 5 | 5 | 1.42 | 0.57 | 43.57 | 1.46 | 0.58 | 44.83 | 1.52 | 0.60 | 46.80 | 1.57 | 0.63 | 48.40 | **1.59** | **0.64** | **49.01** | 1.57 | 0.63 | 48.40 | 1.52 | 0.60 | 46.80 | 1.46 | 0.58 | 44.83 |
| 14 | 5 | 5 | 5 | 1.42 | 0.57 | 43.57 | 1.46 | 0.58 | 44.83 | 1.52 | 0.60 | 46.80 | 1.57 | 0.63 | 48.40 | **1.59** | **0.64** | **49.01** | 1.57 | 0.63 | 48.40 | 1.52 | 0.60 | 46.80 | 1.46 | 0.58 | 44.83 |

Table 4: Post-training dynamics after 315,000 steps in OpenAI highway [21]. Temporal dynamics of DL-based ring attractor with uniform input cues ($K_1 = K_2$). Bold values indicate peak activation at each timestep.

| Step | Cue Ctr | DL ArgMax | DLN1 | | | DLN2 | | | DLN3 | | | DLN4 | | | DLN5 | | | DLN6 | | | DLN7 | | | DLN8 | | |
|---|---|---|---|---|---|---|---|---|---|---|---|---|---|---|---|---|---|---|---|---|---|---|---|---|---|---|
| | | | In | Hid | Out | In | Hid | Out | In | Hid | Out | In | Hid | Out | In | Hid | Out | In | Hid | Out | In | Hid | Out | In | Hid | Out |
| 1 | 2 | 8 | 1.57 | -0.26 | 28.82 | 1.59 | 0.31 | 40.35 | 1.57 | 0.06 | 35.99 | 1.52 | 0.25 | 38.97 | 1.46 | 0.20 | 36.48 | 1.42 | -0.39 | 22.46 | 1.46 | -0.33 | 24.79 | **1.52** | **0.36** | **41.32** |
| 2 | 2 | 2 | 1.57 | -0.39 | 25.92 | **1.59** | **0.09** | **36.94** | 1.57 | -0.28 | 28.48 | 1.52 | -0.07 | 32.04 | 1.46 | -0.10 | 29.82 | 1.42 | -0.66 | 16.62 | 1.46 | -0.64 | 17.97 | 1.52 | -0.26 | 27.89 |
| 3 | 2 | 2 | 1.57 | -0.39 | 25.95 | **1.59** | **0.04** | **35.93** | 1.57 | -0.30 | 27.99 | 1.52 | -0.08 | 31.84 | 1.46 | -0.15 | 28.89 | 1.42 | -0.64 | 17.07 | 1.46 | -0.52 | 20.73 | 1.52 | -0.20 | 29.20 |
| 4 | 2 | 2 | 1.57 | -0.35 | 26.99 | **1.59** | **0.09** | **37.02** | 1.57 | -0.24 | 29.42 | 1.52 | -0.02 | 33.05 | 1.46 | -0.08 | 30.46 | 1.42 | -0.57 | 18.58 | 1.46 | -0.49 | 21.42 | 1.52 | -0.12 | 30.91 |
| 5 | 2 | 2 | 1.57 | -0.35 | 26.86 | **1.59** | **0.10** | **37.18** | 1.57 | -0.23 | 29.44 | 1.52 | -0.02 | 33.15 | 1.46 | -0.07 | 30.53 | 1.42 | -0.58 | 18.29 | 1.46 | -0.50 | 21.11 | 1.52 | -0.12 | 30.87 |
| 6 | 2 | 2 | 1.57 | -0.36 | 26.69 | **1.59** | **0.09** | **36.97** | 1.57 | -0.25 | 29.16 | 1.52 | -0.03 | 32.89 | 1.46 | -0.08 | 30.26 | 1.42 | -0.60 | 18.04 | 1.46 | -0.51 | 20.91 | 1.52 | -0.14 | 30.48 |
| 7 | 5 | 2 | 1.42 | -0.36 | 23.24 | **1.46** | **0.09** | **34.03** | 1.52 | -0.25 | 28.07 | 1.57 | -0.03 | 33.98 | 1.59 | -0.08 | 33.16 | 1.57 | -0.59 | 21.56 | 1.52 | -0.51 | 22.40 | 1.46 | -0.14 | 29.10 |
| 8 | 5 | 4 | 1.42 | -0.33 | 23.82 | 1.46 | 0.05 | 33.28 | 1.52 | -0.25 | 27.92 | **1.57** | **-0.05** | **33.67** | 1.59 | -0.07 | 33.32 | 1.57 | -0.59 | 21.53 | 1.52 | -0.53 | 21.92 | 1.46 | -0.15 | 28.87 |
| 9 | 5 | 5 | 1.42 | -0.34 | 23.70 | 1.46 | 0.04 | 33.09 | 1.52 | -0.25 | 27.98 | 1.57 | -0.06 | 33.31 | **1.59** | **-0.06** | **33.61** | 1.57 | -0.60 | 21.43 | 1.52 | -0.52 | 22.19 | 1.46 | -0.14 | 29.08 |
| 10 | 5 | 5 | 1.42 | -0.34 | 23.75 | 1.46 | 0.04 | 33.09 | 1.52 | -0.25 | 28.04 | 1.57 | -0.06 | 33.31 | **1.59** | **-0.06** | **33.72** | 1.57 | -0.60 | 21.46 | 1.52 | -0.52 | 22.15 | 1.46 | -0.14 | 29.07 |
| 11 | 5 | 5 | 1.42 | -0.34 | 23.74 | 1.46 | 0.04 | 33.08 | 1.52 | -0.25 | 28.04 | 1.57 | -0.06 | 33.32 | **1.59** | **-0.06** | **33.73** | 1.57 | -0.60 | 21.43 | 1.52 | -0.52 | 22.16 | 1.46 | -0.14 | 29.07 |
| 12 | 5 | 5 | 1.42 | -0.34 | 23.74 | 1.46 | 0.04 | 33.07 | 1.52 | -0.25 | 28.04 | 1.57 | -0.06 | 33.31 | **1.59** | **-0.06** | **33.73** | 1.57 | -0.60 | 21.42 | 1.52 | -0.52 | 22.15 | 1.46 | -0.14 | 29.06 |
| 13 | 5 | 5 | 1.42 | -0.34 | 23.74 | 1.46 | 0.04 | 33.07 | 1.52 | -0.25 | 28.04 | 1.57 | -0.06 | 33.31 | **1.59** | **-0.06** | **33.73** | 1.57 | -0.60 | 21.42 | 1.52 | -0.52 | 22.16 | 1.46 | -0.14 | 29.06 |
| 14 | 5 | 5 | 1.42 | -0.34 | 23.74 | 1.46 | 0.04 | 33.07 | 1.52 | -0.25 | 28.04 | 1.57 | -0.06 | 33.31 | **1.59** | **-0.06** | **33.73** | 1.57 | -0.60 | 21.42 | 1.52 | -0.52 | 22.16 | 1.46 | -0.14 | 29.07 |

Table 5: Post-training dynamics after 750,000 steps in OpenAI highway [21]. Uniform input cues ($K_1 = K_2$). Bold values indicate peak activation.

| Step | Cue Ctr | DL ArgMax | DLN1 | | | DLN2 | | | DLN3 | | | DLN4 | | | DLN5 | | | DLN6 | | | DLN7 | | | DLN8 | | |
|---|---|---|---|---|---|---|---|---|---|---|---|---|---|---|---|---|---|---|---|---|---|---|---|---|---|---|
| | | | In | Hid | Out | In | Hid | Out | In | Hid | Out | In | Hid | Out | In | Hid | Out | In | Hid | Out | In | Hid | Out | In | Hid | Out |
| 1 | 2 | 2 | 1.57 | -0.19 | 31.24 | **1.59** | **0.42** | **43.78** | 1.57 | 0.13 | 37.46 | 1.52 | 0.31 | 40.52 | 1.46 | 0.27 | 38.91 | 1.42 | -0.32 | 24.83 | 1.46 | -0.28 | 26.35 | 1.52 | 0.29 | 39.17 |
| 2 | 2 | 2 | 1.57 | -0.31 | 28.47 | **1.59** | **0.15** | **39.26** | 1.57 | -0.21 | 31.02 | 1.52 | -0.02 | 34.81 | 1.46 | -0.05 | 32.37 | 1.42 | -0.58 | 19.14 | 1.46 | -0.56 | 20.52 | 1.52 | -0.19 | 30.46 |
| 3 | 2 | 2 | 1.57 | -0.32 | 28.51 | **1.59** | **0.11** | **38.19** | 1.57 | -0.23 | 30.54 | 1.52 | -0.03 | 34.58 | 1.46 | -0.09 | 31.42 | 1.42 | -0.57 | 19.61 | 1.46 | -0.46 | 23.18 | 1.52 | -0.14 | 31.73 |
| 4 | 2 | 2 | 1.57 | -0.28 | 29.54 | **1.59** | **0.16** | **39.38** | 1.57 | -0.18 | 31.96 | 1.52 | 0.03 | 35.62 | 1.46 | -0.03 | 32.98 | 1.42 | -0.49 | 21.13 | 1.46 | -0.42 | 23.94 | 1.52 | -0.07 | 33.42 |
| 5 | 2 | 2 | 1.57 | -0.28 | 29.39 | **1.59** | **0.17** | **39.56** | 1.57 | -0.17 | 31.98 | 1.52 | 0.04 | 35.73 | 1.46 | -0.02 | 33.06 | 1.42 | -0.51 | 20.82 | 1.46 | -0.43 | 23.61 | 1.52 | -0.06 | 33.38 |
| 6 | 2 | 2 | 1.57 | -0.29 | 29.21 | **1.59** | **0.16** | **39.34** | 1.57 | -0.19 | 31.68 | 1.52 | 0.02 | 35.45 | 1.46 | -0.03 | 32.78 | 1.42 | -0.53 | 20.56 | 1.46 | -0.44 | 23.41 | 1.52 | -0.08 | 32.97 |
| 7 | 5 | 2 | 1.42 | -0.29 | 25.76 | **1.46** | **0.16** | **36.41** | 1.52 | -0.19 | 30.59 | 1.57 | 0.02 | 36.52 | 1.59 | -0.03 | 35.68 | 1.57 | -0.52 | 24.08 | 1.46 | -0.44 | 24.92 | 1.46 | -0.08 | 31.62 |
| 8 | 5 | 4 | 1.42 | -0.26 | 26.34 | 1.46 | 0.12 | 35.64 | 1.52 | -0.19 | 30.43 | **1.57** | **-0.01** | **36.19** | 1.59 | -0.02 | 35.84 | 1.57 | -0.52 | 24.05 | 1.52 | -0.46 | 24.43 | 1.46 | -0.09 | 31.38 |
| 9 | 5 | 5 | 1.42 | -0.27 | 26.21 | 1.46 | 0.11 | 35.43 | 1.52 | -0.19 | 30.49 | 1.57 | -0.02 | 35.49 | **1.59** | **-0.01** | **36.14** | 1.57 | -0.53 | 23.94 | 1.52 | -0.45 | 24.71 | 1.46 | -0.08 | 31.59 |
| 10 | 5 | 5 | 1.42 | -0.27 | 26.27 | 1.46 | 0.11 | 35.43 | 1.52 | -0.19 | 30.56 | 1.57 | -0.02 | 35.82 | **1.59** | **-0.01** | **36.26** | 1.57 | -0.53 | 23.97 | 1.52 | -0.45 | 24.67 | 1.46 | -0.08 | 31.58 |
| 11 | 5 | 5 | 1.42 | -0.27 | 26.26 | 1.46 | 0.11 | 35.42 | 1.52 | -0.19 | 30.56 | 1.57 | -0.02 | 35.83 | **1.59** | **-0.01** | **36.27** | 1.57 | -0.53 | 23.94 | 1.52 | -0.45 | 24.68 | 1.46 | -0.08 | 31.58 |
| 12 | 5 | 5 | 1.42 | -0.27 | 26.26 | 1.46 | 0.11 | 35.41 | 1.52 | -0.19 | 30.56 | 1.57 | -0.02 | 35.82 | **1.59** | **-0.01** | **36.27** | 1.57 | -0.53 | 23.93 | 1.52 | -0.45 | 24.67 | 1.46 | -0.08 | 31.57 |
| 13 | 5 | 5 | 1.42 | -0.27 | 26.26 | 1.46 | 0.11 | 35.41 | 1.52 | -0.19 | 30.56 | 1.57 | -0.02 | 35.82 | **1.59** | **-0.01** | **36.27** | 1.57 | -0.53 | 23.93 | 1.52 | -0.45 | 24.68 | 1.46 | -0.08 | 31.57 |
| 14 | 5 | 5 | 1.42 | -0.27 | 26.26 | 1.46 | 0.11 | 35.41 | 1.52 | -0.19 | 30.56 | 1.57 | -0.02 | 35.82 | **1.59** | **-0.01** | **36.27** | 1.57 | -0.53 | 23.93 | 1.52 | -0.45 | 24.68 | 1.46 | -0.08 | 31.58 |

### A.5.1 Experimental Protocol

We present four experimental tables that track hidden states and outputs across all eight neurons (DLN1-DLN8) in the ring attractor. The experimental setup consisted of an 8-neuron ring configuration with parameters tuned for single stable bump formation: $\tau = 120.0$ (temporal integration constant) and $\beta = 22.0$ (output scaling). The protocol involved an initial Gaussian input centered on neuron 2 for $t \in [1, 6]$, followed by an instantaneous transition to a Gaussian input centered on neuron 5 for $t \geq 7$. This experiment tests the network's ability to maintain stable attractor states and transition smoothly between them, as predicted by theoretical models of ring attractor dynamics [17]. Tables 2 and 3 assess pre-training dynamics under this two-stimulus protocol with asymmetric ($K_2 = 0.1 \cdot K_1$) and matched ($K_1 = K_2$) input amplitudes, respectively. Tables 4 and 5 present post-training dynamics after 315,000 and 750,000 training steps in the OpenAI Highway environment [21], revealing how task-specific learning modifies the ring attractor structure while preserving fundamental attractor properties.

### A.5.2 Preservation of Attractor Dynamics

Table 2 presents the temporal evolution of the DL-based ring attractor under the two-stimulus protocol with asymmetric input amplitudes where $K_2 = 0.1 \cdot K_1$, following the input signal formulation in Eq. 5. Despite the order-of-magnitude difference in input strength, the network transitions between attractor states. During the initial phase (steps 1-6), neuron 2 maintains the highest output values (Out $\approx 55.21$), forming a stable activity bump. Following the input shift at step 7, the argmax tracks positions through intermediate neurons ($2 \to 2 \to 3 \to 4 \to 5$ over steps 7-11), exhibiting the characteristic temporal resistence of ring attractors [42]. By step 12, neuron 5 stabilizes at Output $\approx 5.19$ despite the weaker input signal. Table 3 presents the same protocol with matched input amplitudes ($K_1 = K_2$). The dynamics exhibit three characteristic phases consistent with CANN behavior [4]. During the initial attractor formation phase ($t \in [1, 6]$), the network maintains a stable activity bump centered at neuron 2, with the argmax operation consistently identifying the correct peak location. The hidden state magnitudes remain comparable between the two stable phases (DLN2 Hidden $\approx 0.77$ at step 1 vs. DLN5 Hidden $\approx 0.64$ at step 12), confirming that the asymmetries observed in Table 2 arise from input differences rather than structural biases. Following the input shift at $t = 7$, the network demonstrates smooth state transition dynamics ($t \in [7, 11]$), where the activity bump migrates from neuron 2 to neuron 5. The network subsequently converges to a new stable attractor state ($t \in [12, 14]$), settling into a configuration centered at neuron 5. This temporal lag reflects the $\tau = 120.0$ integration constant and should demonstrate the network's ability to maintain stable representations while integrating new information.

Tables 4 and 5 present post-training dynamics after 315,000 and 750,000 training steps in the Highway environment, respectively. In Table 4, following the input shift at step 7, the peak transitions from neuron 2 (step 2, Output $\approx 36.94$) through neuron 4 (step 8, Output $\approx 33.67$) to neuron 5 (steps 9-14, Output $\approx 33.73$). The hidden states include negative values (e.g., DLN5 Hidden $= -0.06$ at convergence), indicating that task-specific learning has modified internal dynamics while maintaining the output behavior necessary for action selection. The attractor state of a neuron is determined by the aggregate influence of all incoming connections through the hidden-to-hidden weights. As the hidden state undergoes transformation through $y_t = f(W_{hh} \cdot h_t + b_y)$, negative or non-maximal hidden values can produce the appropriate output behavior after passing through the tanh activation and scaling parameter $\beta$. Table 5 shows stabilization of these learned dynamics at 750k steps, with similar structural features confirming that the adaptations represent stable task-specific encoding rather than transient learning dynamics. Neuron 6, corresponding to the "move backwards" action in Highway, consistently shows lower activation values across both checkpoints (Output $\approx 22.46$ at step 1 in Table 4), reflecting the penalty associated with this action.

### A.5.3 Analysis

The preservation of attractor dynamics post-training aligns with findings in biological systems, where synaptic plasticity modifies connection strengths while maintaining functional circuit properties [37, 24]. We may infer from these experimental results that our DL implementation preserves essential CANN dynamics while enabling task-specific adaptation through learning. The maintained bump migration and state persistence properties, combined with the evidence of spatial representations provided in the ablation studies in Section A.4, indicate that performance improvements stem from the ring attractor's inherent spatial encoding capabilities and attractor states rather than merely adding recurrent connections. The preservation of attractor-like dynamics post-training, despite substantial weight evolution away from the initial Gaussian profile, reveals several computational mechanisms at work. The architecture maintains spatial awareness through the explicit ring topology: the circular arrangement of neurons preserves spatial relationships between actions throughout training, as evidenced by the natural preservation of distance-dependent weights in the input-to-

hidden connections and the smooth bump migration patterns observed across all experimental conditions in Section A.4. The ablation studies demonstrate that removing the ring structure causes performance to drop below baseline levels, confirming that spatial encoding provides functionally significant advantages over standard RNNs and feed-forward networks. Simultaneously, the learnable $\tau$ parameter introduces temporal filtering dynamics that serve distinct functions across training phases. During early training, the results suggest that the architecture also functions as a low-pass filter, where $\tau$ creates momentum that reduces jitter in action exploration and enables the agent to reach distant state-space regions more efficiently.

### A.6 Deep Learning Ring Attractor Recurrent Neural Network Modeling

As seen before, excitatory neurons are organized in a circular pattern, with connection weights between neurons determined by a distance-weighted function mimicking the synaptic connection of biological neurons. Fig. 9 illustrates the structured connectivity of the RNN, which mimics the circular topology of biological ring attractors.

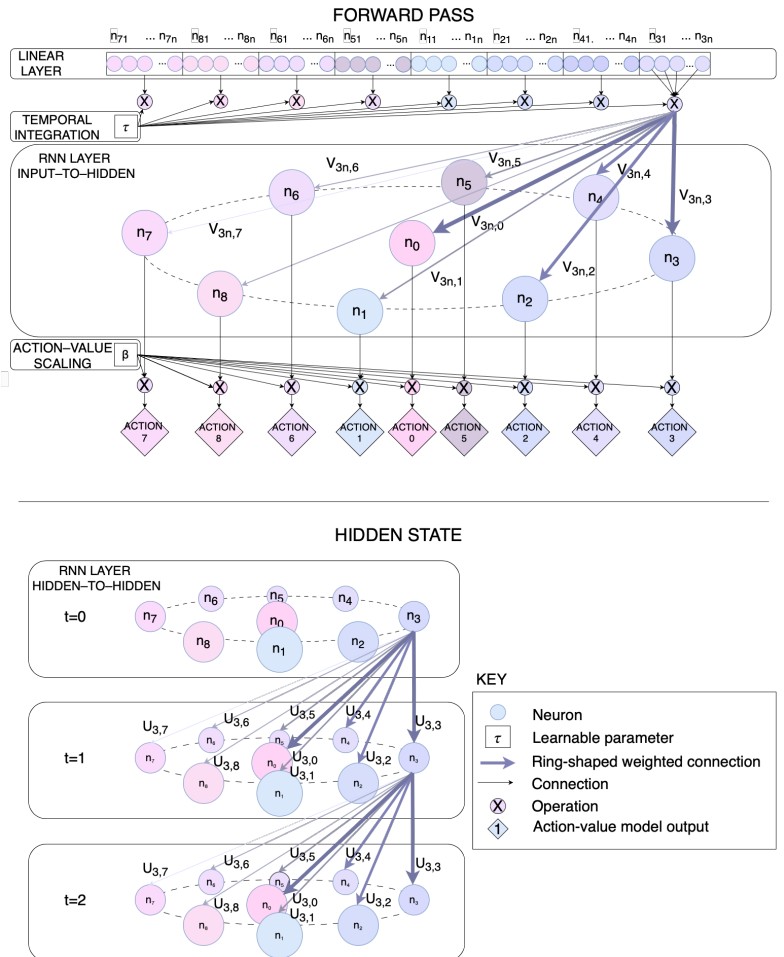

Figure 9: RNN modeling ring attractor synaptic connections: The top panel shows the forward pass (input-to-hidden) as the agent output layer. The bottom panel depicts the hidden-to-hidden recurrent connection between inference time steps. Weighted connections are illustrated for the sample neuron $n_3$.

### A.7 Deep Learning Ring Attractor Model Implementation Details

As presented in Section 3.2, both the input-to-hidden connections $V(s)$ and hidden-to-hidden connections $U(v)$ are constructed using the distance-dependent weight functions defined in Eq. (12).

These equations establish the circular topology of the ring attractor and determine how information flows through the network. However, a special case arises when dealing with neutral actions in certain Atari games, requiring a modification to the standard distance function.

### A.7.1 Neutral Inhibitory Action Implementation

For games in the Atari benchmark with neutral actions (like 'no-op'), the ring attractor maintains its circular structure with a neutral action positioned centrally. This central position creates equal connections of strength 1 to all other actions in the ring, as if it were a direct neighbor to each action simultaneously. The distance between the neutral action $n$ and any other action $m$ is fixed at 1:

$$d(m,n) = \begin{cases} 1, & \text{if } m \text{ or } n \text{ is the neutral action} \\ d(m,n), & \text{otherwise,} \end{cases} \tag{25}$$

where $d(m,n)$ remains as defined in Eq. (12) for all other action pairs. The weight matrices $w_{I \to H}$ and $w_{H \to H}$ maintain the same exponential decay based on this distance function.

For example, in games like Seaquest or Asterix, this central positioning means the "no-op" action has consistent, strong connections to all directional actions. This arrangement preserves the spatial relationships between directional actions while ensuring the neutral action remains equally accessible from any game state. The constant distance of 1 to all other actions makes transitioning to or from the neutral action as natural as moving between adjacent directional actions in the ring.

### A.7.2 Deep Learning Double Ring Attractor Equations

For a double ring configuration in our DL implementation as presented in the experiments, Section 4.2, the weighted connections are defined as follows:

Let $\mathbf{V}^{double} \in \mathbb{R}^{2N \times 2M}$ be the complete input-to-hidden weight matrix for both rings, where $N$ is the number of output neurons per ring and $M$ is the number of input features per ring. The matrix is structured as:

$$\mathbf{V}^{double} = \begin{bmatrix} \mathbf{V}_{11} & \kappa \mathbf{V}_{12} \\ \kappa \mathbf{V}_{21} & \mathbf{V}_{22} \end{bmatrix}, \tag{26}$$

where $\mathbf{V}11 = \mathbf{V}22 = \mathbf{V}12 = \mathbf{V}21$, $\kappa = 0.1$ is the cross-coupling learnable parameter initialised to 0.1. This allows the network to learn the optimal strength of interaction between the two rings during training.

Developing from Eq. (12), each ring maintains identical connectivity patterns, preserving the spatial relationships of their respective action dimensions, each submatrix $\mathbf{V}_{ij}$ represents:

$$[\mathbf{V}_{ii}]_{m,n} = \frac{1}{\tau} \Phi_\theta(s)^T e^{-d(m,n)/\lambda}, \tag{27}$$

where $d(m,n)$ is defined as per the forward pass weighted connections in Eq. (12).

Similarly, let $\mathbf{U}^{double} \in \mathbb{R}^{2N \times 2N}$ be the complete hidden-to-hidden weight matrix:

$$\mathbf{U}^{double} = \begin{bmatrix} \mathbf{U}_{11} & \kappa \mathbf{U}_{12} \\ \kappa \mathbf{U}_{21} & \mathbf{U}_{22} \end{bmatrix} \tag{28}$$

For primary connections ($\mathbf{U}_{11}$ and $\mathbf{U}_{22}$):

$$[\mathbf{U}_{ii}]_{m,n} = h(v)^T e^{-d(m,n)/\lambda}, \tag{29}$$

where $d(m,n)$ is defined as per the hidden state weighted connections in Eq. (12).

The complete forward pass for both rings is given by:

$$Q(s,a) = \beta \tanh \left( \begin{bmatrix} \frac{1}{\tau} \Phi_\theta(s_t)^T \mathbf{V}_{11} + h_{t-1}(v)^T \mathbf{U}_{11} & \frac{\kappa}{\tau} \Phi_\theta(s_t)^T \mathbf{V}_{12} + \kappa h_{t-1}(v)^T \mathbf{U}_{12} \\ \frac{\kappa}{\tau} \Phi_\theta(s_t)^T \mathbf{V}_{21} + \kappa h_{t-1}(v)^T \mathbf{U}_{21} & \frac{1}{\tau} \Phi_\theta(s_t)^T \mathbf{V}_{22} + h_{t-1}(v)^T \mathbf{U}_{22} \end{bmatrix} \right), \tag{30}$$

where $\tau$ is the learnable time constant; $\beta$ is the learnable scaling factor; $\Phi_\theta(s_t)$ is the feature representation of state $s_t$; $h_{t-1}(v)$ is the previous hidden state; and $\kappa = 0.1$ is the coupling strength between rings.

The cross-coupling matrices ($\kappa\mathbf{V}12$, $\kappa\mathbf{V}21$, $\kappa\mathbf{U}12$, and $\kappa\mathbf{U}21$) maintain a circular topology similar to the individual rings. A neuron at a particular position in the first ring connects most strongly to the neuron at the corresponding position in the second ring, with connection strength decreasing based on circular distance. This structured cross-coupling preserves spatial alignment between the two action dimensions while allowing semi-independent operation through the learnable coupling factor $\kappa$. The final output provides action-values for both action dimensions simultaneously, preserving spatial relationships within each ring while allowing weak coupling between them.

### A.7.3    Extension to N Ring Configurations

The double ring implementation extends to $R$ rings through a block matrix structure, where each ring encodes a distinct action dimension. For $R$ rings, the architecture uses block matrices $V_{\text{multi}} \in \mathbb{R}^{RN \times RM}$ and $U_{\text{multi}} \in \mathbb{R}^{RN \times RN}$, with diagonal blocks preserving individual ring dynamics and off-diagonal blocks handling cross-ring interactions via coupling parameter $\kappa$, as seen in Section A.7.2. While computational complexity scales as $O(R^2)$, selective coupling between only related dimensions creates a sparse structure with effective $O(R)$ complexity. This makes the approach viable for complex action spaces where actions decompose into multiple semi-independent planes, for example, in games that combine movement, combat, and resource-management dimensions.

### A.8    Models and Environments Implementation

Table 6: Implementation details for ring attractor architectures across environments. The table shows the environment (Env); ring configuration (Ring); number of actions or continuous 1D action space (Actions); inhibitory neuron placed equidistant to other neurons for "no action" term (Inhib); whether uncertainty estimation is used (Uncert); the implemented model (Model); and type of Neural Network used (Type).

| Game Environment | Ring | Configuration Actions | Inhib. | Uncert. | Implementation Model | Type |
|---|---|---|---|---|---|---|
| Highway | Single | Continuous | No | Yes | BDQNRA-UA | CTRNN |
| Mario Bros | Single | 8 | No | Yes | BDQNRA-UA | CTRNN |
| Highway | Single | 8 | No | No | DDQNRA | DL-RNN |
| Mario Bros | Single | 8 | No | No | DDQNRA | DL-RNN |
| Alien | Double | 18 | Yes | No | EffZeroRA | DL-RNN |
| Asterix | Single | 9 | Yes | No | EffZeroRA | DL-RNN |
| Bank Heist | Double | 18 | Yes | No | EffZeroRA | DL-RNN |
| BattleZone | Double | 18 | Yes | No | EffZeroRA | DL-RNN |
| Boxing | Double | 18 | Yes | No | EffZeroRA | DL-RNN |
| Chopper C. | Double | 18 | Yes | No | EffZeroRA | DL-RNN |
| Crazy Climber | Single | 9 | Yes | No | EffZeroRA | DL-RNN |
| Freeway | Double | 18 | Yes | No | EffZeroRA | DL-RNN |
| Frostbite | Double | 18 | Yes | No | EffZeroRA | DL-RNN |
| Gopher | Double | 18 | Yes | No | EffZeroRA | DL-RNN |
| Hero | Double | 18 | Yes | No | EffZeroRA | DL-RNN |
| Jamesbond | Double | 18 | Yes | No | EffZeroRA | DL-RNN |
| Kangaroo | Double | 18 | Yes | No | EffZeroRA | DL-RNN |
| Krull | Double | 18 | Yes | No | EffZeroRA | DL-RNN |
| Kung Fu M. | Double | 18 | Yes | No | EffZeroRA | DL-RNN |
| Ms Pacman | Single | 9 | Yes | No | EffZeroRA | DL-RNN |
| Private Eye | Double | 18 | Yes | No | EffZeroRA | DL-RNN |
| Road Runner | Double | 18 | Yes | No | EffZeroRA | DL-RNN |
| Seaquest | Double | 18 | Yes | No | EffZeroRA | DL-RNN |

We provide implementation details for both our models and the tested environments. For model implementations, EffZeroRA was applied across the Atari benchmark suite, while BDQNRA-UA and DDQNRA were specifically implemented for Highway and Mario Bros. Table 6 details the configuration of action spaces and ring architectures for each environment. The environments required different ring configurations based on their control schemes, ranging from single-ring implementations for basic movement to double-ring setups for more complex action spaces that combine movement and specialized actions. Each ring topology was designed to preserve the natural

relationships between actions, with central inhibitory actions included where appropriate as "no action".

### A.8.1 Computational Resources

For the Atari 100K benchmark [5] experiments, we utilized a high-performance computing cluster equipped with 10 NVIDIA A100 GPUs (each with 80 GB memory), 512 GB of RAM, and 128 Intel Xeon CPU cores running at 2.4 GHz. The cluster environment enabled parallel execution of experiments across multiple seeds and games simultaneously, with training times ranging from 8 to 10 hours per game, depending on complexity.

For the other environments, Highway [21] and Super Mario Bros [15], we employed a local workstation with an Intel Xeon processor (28 cores, 2.1 GHz), 125 GB RAM, and dual NVIDIA RTX A5000 GPUs (total 48 GB combined memory). Training on the local machine typically required 6-12 hours per environment, with the CTRNN implementation incurring additional computational overhead as noted in Section 4.1. All experiments were conducted using PyTorch 1.12 with CUDA 11.6. The total computational cost is estimated at approximately 8,000 GPU-hours across all experiments and ablation studies.

