# OpenReview forum: "Spatial-Aware Decision-Making with Ring Attractors in Reinforcement Learning Systems"
_NeurIPS.cc/2025/Conference — NeurIPS 2025 poster_

### Official Review · Reviewer_nzPF · 2025-07-03

**Clarity:** 2
**Significance:** 3
**Originality:** 3
**Rating:** 3
**Confidence:** 4

**Summary:**

This paper proposed a method to extend existing discrete reinforcement learning models with a ring attractor and a CANN-like structure. The method improves the performance of those models in the Atari 100k benchmark.

**Questions:**

1. In line 172, why $K_i=Q(s, a)$?
2. In Equation (5), why does the parameter $s$ exist on the right-hand but not on the left-hand?
3. How do you sample in line 257?
4.  The time constant $tau$ seems very quick. What is the final $tau$?
5. How does $tau$ participate in Equation 12?

**Ethical Concerns:**

["NO or VERY MINOR ethics concerns only"]

**Final Justification:**

During the rebuttal stage, my questions in the discussion focused on examining whether the deep learning implementation of the CANN in this paper preserves the core properties of the original CANN, and whether the model truly exhibits "spatial awareness."  The authors' responses have helped me better understand the actual role of the "ring attractor" mechanism in this work and the underlying reasons for the observed performance improvements.

I appreciate the technical contribution, but I also would like to emphasize that the explanation of the core mechanism in the paper appears to be inaccurate or misleading. The model integrates equations inspired by CANNs into an RNN layer and attributes its effectiveness to "Spatial-Aware Decision-Making." However, based on the discussion and evidence provided, this does not seem to be the true reason for the performance gains.

According to the clarifications, the layer primarily functions as a low-pass filter during the early stages of training, which reduces jitters in action exploration and allows the agent to explore more distant regions more quickly. In the later stages, the layer behaves more like a general RNN with temporal integration, which helps the network better fit the Q-values. These dynamics, while useful, are not reflective of spatial representation or attractor dynamics in the conventional sense of CANNs.

I recommend that the authors revise the manuscript to more precisely describe the role of the RNN layer, avoiding potentially misleading explanations regarding spatial awareness. A more accurate characterization would strengthen the clarity and integrity of the paper and improve its value to the community.

I believe this paper has the potential to be accepted, provided the authors revise the manuscript to more accurately and clearly explain the underlying mechanisms. If this was a journal review, I would recommend a major revision, with the intention of recommending acceptance depending on the quality of the revision. However, given the current version and the fact that the authors have not yet fully recognized or addressed this issue, I am maintaining my current score.

**Limitations:**

This paper does not discuss this part, but I don't think it has a potential negative societal impact.

**Paper Formatting Concerns:**

No.

**Quality:**

2

**Strengths And Weaknesses:**

Strengths:

The idea to extend existing discrete reinforcement learning models with ring attractors is innovative, and the result suggests a performance improvement.

Weakness:

The method is not well-explained, and the math is hard to follow due to inconsistencies and notations. Although the part about the ring attractors is introduced, the exact way to extend existing models, such as those shown in the experiments, is not clearly stated. The functionality of the network that forms the ring attractor is not clear. According to the appendix, the deep learning ring attractor functions like an extra RNN layer in addition to existing models. The dynamics of the ring attractor, which is its most important feature, is not supported or demonstrated in the experiments or the appendix.

Minor issues:
1. The differential notation is noted as $d$ in the paper. It should not be italic, otherwise it is mixed with $d$ in line 159.
2. Line  159 and line 160: $d^2_(m,n)$ refers to difference concept. So as in Equation 12.
3. max and argmax should not be italic.
4. Equations in Figure 1 are not clear. For example, what is $argmax(dv/dt)$? Is it explained in the text?
5. $theta$ is used for different concepts, including parameters of different models, but is not distinguished. It is really confusing.

I also suggest that the authors should prove that the performance of the baseline models cannot result in a better performance by adding a typical RNN or FNN layer, to discuss the possibility that the performance improvement is due to more parameters.

---

> ### Author Rebuttal · Authors · 2025-07-28
>
> We would like to deeply thank the reviewer for their detailed feedback on our work, especially on the methodology front.
>
> ## Regarding the Strengths of the Article
>
> Briefly, we would like to point out that we have also conducted experiments with a 1D continuous action space, Fig. 2. The intent is to showcase the ability of the ring attractor to handle both continuous action spaces and discrete ones, the latter ones being used in well-established benchmarks such as Atari.
>
> ## Regarding the Extension of Existing Models in the Article
>
> We believe Section 3.2.1 provides guidance for practitioners. Algorithm 1 offers step-by-step dynamics loop for the CTRNN implementation with Eq. 5, 6, 7, 8, and 11 being the key equations to map fundamentals, while Section 3.2.1 presents the mathematical formulations for the DL architecture. Additionally, Appendix A.6 goes through the integration of multi-ring architectures with the DRL approach when in need to cover higher dimensional spaces, and Appendix A.7 walks the reader through the particular implementation for each environment evaluated in the article.
>
> For further insights, "plug and play" implementation and a follow-up on how this is being implemented into a fully functioning algorithm, we refer to our available codebase in the link in the abstract where we show multiple implementations for the CTRNN implementation and DL-based implementation of the ring attractor.
>
> **Action:** However, we acknowledge that this information might benefit from being more prominently featured in the article. We will work on adding distilled pseudo-code from the implementation in the appendix.
>
> ## Concerning the Demonstration of Ring Attractor Dynamics
>
> We believe our paper provides substantial evidence for the importance of attractor dynamics through a set of sections in the article. Section 3.1 gives the differential equations for the excitatory and inhibitory populations Eq. 6, 7 that generate the bump-of-activity characteristic of a ring attractor. Algorithm 1 shows that on every decision (action selection) step, we iterate these equations to reach attractor state, ensuring the network actually settles before an action is read out.
>
> Most importantly, we believe several ablation studies deliberately disrupt the dynamics while keeping all other code paths unchanged as data points to support our statements empircally:
>
> - **CTRNN**: BDQNRA-RM randomly permutes actions around the ring (scrambling distance-dependent interactions). This results in Fig. 6, showing a glorified forward pass that doesn't alter baseline model performance.
> - **DL-RNN**: DL-no-ring replaces the exponential decay distance weight matrix with a standard initialization matrix (eliminating spatially-structured recurrence), causing again performance in line with baseline Fig. 6. Because the only difference from the full model is the absence of proper ring interactions, we believe these results constitute strong empirical support that the settling dynamics are present and causally useful.
>
> Appendix A.4.3 tracks the learned weight matrices, showing that forward (input→ring) weights keep their spatial profile throughout training, while recurrent weights adapt task-specific asymmetries, Fig. 7, when leaving the DL-based ring attractor distance-based weights trainable. The persistence of this profile indicates that the network consistently reaches and learns to exploit the intended attractor regime.
>
> ## On the Characterization and Ablation of an RNN Layer
>
> While our DL implementation does utilize RNN architecture, it fundamentally differs from standard recurrent layers through its structured single/multi-ring circular connectivity and distance-dependent weights (Eq. 12). It modulates the input and hidden state through two parameters: the learnable time constant τ and the distance decay parameter λ. The key innovation lies not in adding computational layers, as shown in the ablation studies in Appendix A.4, but in encoding spatial relationships through biologically-inspired ring topology.
>
> This structured approach preserves action relationships and enables the spatial encoding capabilities that drive our performance improvements. As mentioned above, Fig. 6 shows how the RL agent performs when distance-dependent weights are removed and the weights are set back to trainable. Effectively turning the ring attractor layer back to a standard RNN, when doing so, the DRL agent shows similar or slightly lower performance compared to baseline.
>
> ## Addressing Formatting and Notation Issues
>
> ### 1. Differential Equation Notation
> In an effort to improve readability, we will remove italics from the differential equations to set them apart from the distance-based variable.
>
> **Action**: Use `\mathrm{}` to remove italics from differential notation in equations across the article.
>
> ### 2. Distance Function Clarification
> Regarding lines 159-160 and Eq. 12, we respectfully clarify that d(m,n) refers to the same concept throughout: The discrete distance based on the number of neurons between two interacting neurons. The different formulations (`|m-n|` vs. `min(|m-n|, N-|m-n|)`) account for linear vs. circular distance calculations within the ring topology, both representing neuron-to-neuron distance.
>
> ### 3. Mathematical Operators
> We agree with the reviewer and thank them for the feedback. Argmax as operator should not be in italics.
>
> **Action**: Convert Argmax operation to roman format.
>
> ### 4. Differential Equations vs. Argmax
> The differential equations (dv/dt) describe how neuron activations evolve until the ring attractor reaches a stable state, while argmax(V) operates on the final settled activation values V = [v1, v2, ..., vN] to select the action corresponding to the most active neuron, Eq. 8. The dynamics (dv/dt) drive the system to convergence, and argmax(V) reads out the final decision.
>
> **Action**: In order to simplify understanding throughout the article, we will assume that Fig. 1 is at the final attractor (i.e., settled state). This will enable to swap argmax(dv/dt) for argmax(V) and point to Eq. 8, so it's easier for the reader to follow.
>
> ### 5. Parameter Notation Consistency
> θ consistently denotes the learnable parameters of the DRL agent across the different methodologies and the appendix.
>
> **Action**: We will reserve θ for the full parameter set (e.g., the neural network in Section 3.1.3) and use subscripts for specific subsets within the DRL agent.
>
> ## Specific Questions
>
> ### Question 1: In line 172, why Ki = Q(s,a)?
>
> In the original ring attractor formulation (Eq. 1), Ki represents the magnitude or strength of a given input signal i. By setting Ki = Q(s,a), we establish that the magnitude of a given signal is given by a state-action pair for a given action. Effectively this means we map input signals (i) to actions in action space (a), integrating the ring attractor as policy from Eq. 1 in Eq. 5. We will add a remark to highlight the substitution of input signal (i) for actions (a) and we want to thank the reviewer for the point.
>
> **Action**: We will signal this substitution at the beginning of Section 3.1.2 with the following sentence:
> > "To integrate the ring attractor with RL, we reformulate the input signals by mapping Ki to the Q-value Q(s,a), effectively substituting input signal index i with actions a∈A."
>
> ### Question 2: In Eq. (5), why does parameter s exist on the right-hand but not on the left-hand?
>
> We originally used xn(Q) for readability since the summation ∑(a=1 to A) Q(s,a) makes the dependence on specific Q-values explicit.
>
> **Action**: We will revise to xn(Q(s,a)) for notational consistency throughout Eq. 5.
>
> ### Question 3: How do you sample in line 257?
>
> The sampling in line 257 refers to Thompson Sampling within an established Bayesian Linear Regression approach from the referred work by Kamyar Azizzadenesheli, Emma Brunskill, and Animashree Anandkumar, line 227. "Efficient exploration through bayesian deep q-networks." There is also a practical implementation of this method in the codebase linked in the abstract and we would be happy to explain in detail how this method works with our approach if there are any further questions.
>
> ### Question 4: The time constant τ seems very quick. What is the final τ?
>
> In our CTRNN implementation, we set τ = Δt where Δt represents the discrete time step size. For our experiments, we used τ = 1e-3 s, with a simulation time of 0.05 s. This gives a total of 50 iterations to reach attractor state. These values were chosen empirically to optimize the performance of the CTRNN model, and were preserved across experiments (environments, seeds and ablation studies).
>
> **Action**: Add simulation time integration parameters to pseudocode in Algorithm 1.
>
> ### Question 5: How does τ participate in Eq. 12?
>
> In Eq. (12), τ appears in the input signal computation V(s)m,n = (1/τ)Φθ(s)T wI→H{m,n}. Here, τ acts as a scaling factor that controls the contribution of the input features to the hidden state. A smaller τ makes the network more responsive to immediate inputs, while a larger τ creates more integration over time. This is a learnable parameter allowing the network to discover the optimal balance between responsiveness and temporal integration for each specific task.
>
> **Action**: We will expand the existing sentence in line 297 to:
>
> > "Here, τ acts as a learnable scaling factor that controls the balance between immediate input responsiveness and temporal integration within the ring attractor dynamics, with smaller values increasing sensitivity to current inputs and enabling faster adaptation, while larger values promote stability and smoother transitions between attractor states by integrating information over longer time horizons."
>
> We hope these clarifications address the technical concerns raised and demonstrate our commitment to improve the quality of our article.

---

> > ### Comment · Reviewer_nzPF · 2025-08-04
> >
> > Thank you for your detailed response and highlighting the sections. I carefully read all the sections of this paper, which are in detail, but still missing some aspects. Let's try to make the discussion focused.
> > 1. CANN is described as a set of differential equations and solved by numerical integration. Although Figure 9 mentioned temporal interation,  "RNN version" lacks details about how the integration happens, or why these two versions are equivalent.
> > 2. The dynamics of CANN states, which are the firing rate of the hidden neurons, can be presented as curves in time.  Could you present the curves of both versions as a table here? It helps to compare their differences and confirm that the key dynamics of CANN are preserved in the RNN version. Showing the dynamics of the RNN version is the key to proving that the improvement of model performance is benefited by the ring attractor instead of one more RNN layer.

---

> > > ### Author Response · Authors · 2025-08-06
> > >
> > > We thank the reviewer for engaging with our article. We respond to the points below concisely due to space limitations:
> > >
> > > Q1. Temporal Integration Parameter τ refers specifically to the learnable parameter τ inherited from the CTRNN ring attractor implementation, where τ is the temporal integration constant. In Section 3.2.1, it effectively serves as a learnable parameter controlling the previous layer's contribution.
> > >
> > > The equivalence of the DL-based implementation does not rely on precisely replicating the CTRNN architecture but preserves key benefits for RL: explicit action space mapping, action relationships, and faster learning rates. This follows the path of other structures adapted from bio-inspired principles such as perceptrons, ConvNets, or Attention Mechanisms.
> > > Our RL benefit claims are supported by ablation studies. Appendix A.4.3 demonstrates preservation of spatial relationships, showing optimal ring distribution during training when neuron weights remain trainable. Critically, A.4.2 shows removing the ring structure, converting to standard RNN as the reviewer mentions, causes performance drops below baseline levels.
> > >
> > > Q2. We clarify that neither model (CTRNN and DL/RNN) represents true firing rates. CTRNN models represent continuous activation levels, not actual firing rates as in Spiking Neural Networks (SNNs). We acknowledge that while CTRNNs don't model true firing rates, they are interpreted as rate-based models where continuous activations approximate membrane potentials over time.
> > >
> > > As CANN dynamics evidence, we provide hidden state activations from our DL-based model evolution through a controlled two-stimulus protocol designed to demonstrate attractor dynamics. The experimental setup involves a ring with 8 neurons where we maintained DL architecture integrity while tuning parameters to accentuate single stable bump formation: τ=120.0 (slower input drive dynamics) and β=22.0 (moderate output scaling). An initial Gaussian input centered on neuron 2 is followed by instantaneous transition to a second Gaussian input centered on neuron 5 at step 7. Argmax compares bump location between CTRNN and DL-based models. The neuronal response profiles suggest characteristic smooth bump migration between spatial locations from neuron 2 to neuron 5, with hidden state activations providing evidence of attractor state dynamics evolving smoothly between time steps 7 and 15 to accommodate the new input signal.
> > >
> > > |Step|CueCenter|CTRNNArgMax|DLArgMax|DLN1Input|DLN1Hidden|DLN1Output|DLN2Input|DLN2Hidden|DLN2Output|DLN3Input|DLN3Hidden|DLN3Output|DLN4Input|DLN4Hidden|DLN4Output|DLN5Input|DLN5Hidden|DLN5Output|DLN6Input|DLN6Hidden|DLN6Output|DLN7Input|DLN7Hidden|DLN7Output|DLN8Input|DLN8Hidden|DLN8Output|
> > > |---|---|---|---|---|---|---|---|---|---|---|---|---|---|---|---|---|---|---|---|---|---|---|---|---|---|---|---|
> > > |1|2|2|2|1.63|0.75|52.28|**1.73**|**0.78**|**55.21**|1.63|0.75|52.28|1.38|0.67|44.98|1.09|0.58|36.86|0.93|0.55|32.47|1.09|0.58|36.86|1.38|0.67|44.98|
> > > |2|2|2|2|1.63|0.82|53.83|**1.73**|**0.86**|**56.97**|1.63|0.82|53.83|1.38|0.72|46.02|1.09|0.61|37.38|0.93|0.56|32.75|1.09|0.61|37.38|1.38|0.72|46.02|
> > > |3|2|2|2|1.63|0.71|51.50|**1.73**|**0.75**|**54.70**|1.63|0.71|51.50|1.38|0.60|43.54|1.09|0.49|34.74|0.93|0.44|30.05|1.09|0.49|34.74|1.38|0.60|43.54|
> > > |4|2|2|2|1.63|0.65|50.12|**1.73**|**0.69**|**53.34**|1.63|0.65|50.12|1.38|0.54|42.11|1.09|0.42|33.27|0.93|0.37|28.56|1.09|0.42|33.27|1.38|0.54|42.11|
> > > |5|2|2|2|1.63|0.64|49.94|**1.73**|**0.68**|**53.17**|1.63|0.64|49.94|1.38|0.53|41.92|1.09|0.41|33.06|0.93|0.36|28.35|1.09|0.41|33.06|1.38|0.53|41.92|
> > > |6|2|2|2|1.63|0.65|50.14|**1.73**|**0.69**|**53.37**|1.63|0.65|50.14|1.38|0.54|42.12|1.09|0.42|33.26|0.93|0.37|28.55|1.09|0.42|33.26|1.38|0.54|42.12|
> > > |7|5|2|2|0.14|0.65|17.51|**0.15**|**0.70**|**18.60**|0.15|0.65|17.75|0.16|0.54|15.42|0.16|0.43|12.89|0.16|0.37|11.68|0.15|0.43|12.74|0.15|0.54|15.17|
> > > |8|5|2|2|0.14|-0.02|2.72|**0.15**|**-0.00**|**3.13**|0.15|-0.02|3.02|0.16|-0.05|2.43|0.16|-0.08|1.75|0.16|-0.10|1.37|0.15|-0.08|1.55|0.15|-0.05|2.11|
> > > |9|5|3|3|0.14|-0.09|1.24|0.15|-0.08|1.46|**0.15**|**-0.08**|**1.56**|0.16|-0.09|1.50|0.16|-0.10|1.34|0.16|-0.10|1.19|0.15|-0.10|1.13|0.15|-0.09|1.15|
> > > |10|5|3|4|0.14|0.01|3.28|0.15|0.01|3.43|0.15|0.01|3.60|**0.16**|**0.01**|**3.70**|0.16|0.01|3.70|0.16|0.01|3.61|0.15|0.01|3.48|0.15|0.01|3.34|
> > > |11|5|4|5|0.14|0.06|4.50|0.15|0.07|4.64|0.15|0.07|4.82|0.16|0.07|4.97|**0.16**|**0.07**|**5.01**|0.16|0.07|4.94|0.15|0.07|4.79|0.15|0.06|4.61|
> > > |12|5|5|5|0.14|0.07|4.66|0.15|0.07|4.79|0.15|0.07|4.98|0.16|0.08|5.14|**0.16**|**0.08**|**5.19**|0.16|0.08|5.13|0.15|0.07|4.97|0.15|0.07|4.78|
> > > |13|5|5|5|0.14|0.06|4.48|0.15|0.06|4.61|0.15|0.07|4.80|0.16|0.07|4.96|**0.16**|**0.07**|**5.02**|0.16|0.07|4.96|0.15|0.07|4.80|0.15|0.06|4.60|
> > > |14|5|5|5|0.14|0.06|4.36|0.15|0.06|4.48|0.15|0.06|4.68|0.16|0.06|4.84|**0.16**|**0.06**|**4.90**|0.16|0.06|4.84|0.15|0.06|4.68|0.15|0.06|4.48|

---

> > > > ### Comment · Reviewer_nzPF · 2025-08-06
> > > >
> > > > Thank you for the reply. The table provides important details.
> > > >
> > > > 1. Why the DLN2Hidden and DLN2Output at the inital steps are ten times larger than DLN5Hidden and DLN5Ouput at the end stepss, respectively? Is it because the former's input is ten times the latter's input? Can you keep these inputs at a similar scale for a controlled comparison?
> > > >
> > > > 2. Can you provide similar tables of your DLN CANN models after training in the RL tasks?
> > > >
> > > > Thanks

---

> ### Author Response · Authors · 2025-08-06
>
> We thank the reviewer for acknowledging the additions and validating our approach.
>
> Q1. Yes, that is correct. The experiement has been designed with two cues having identical properties, except for their amplitudes to demonstrate robustness under varying cue/input sizes. Below the same experiment using two identical input cues at Step 1 and Step 7.
>
> |Step|CueCenter|CTRNNArgMax|DLArgMax|DLN1Input|DLN1Hidden|DLN1Output|DLN2Input|DLN2Hidden|DLN2Output|DLN3Input|DLN3Hidden|DLN3Output|DLN4Input|DLN4Hidden|DLN4Output|DLN5Input|DLN5Hidden|DLN5Output|DLN6Input|DLN6Hidden|DLN6Output|DLN7Input|DLN7Hidden|DLN7Output|DLN8Input|DLN8Hidden|DLN8Output|
> |---|---|---|---|---|---|---|---|---|---|---|---|---|---|---|---|---|---|---|---|---|---|---|---|---|---|---|---|
> |1|2|2|2|1.57|0.76|51.29|**1.59**|**0.77**|**51.84**|1.57|0.76|51.29|1.52|0.74|49.85|1.46|0.72|48.04|1.42|0.72|46.85|1.46|0.72|48.04|1.52|0.74|49.85|
> |2|2|2|2|1.57|0.82|52.55|**1.59**|**0.82**|**53.15**|1.57|0.82|52.55|1.52|0.80|51.00|1.46|0.77|49.08|1.42|0.76|47.84|1.46|0.77|49.08|1.52|0.80|51.00|
> |3|2|2|2|1.57|0.69|49.82|**1.59**|**0.70**|**50.43**|1.57|0.69|49.82|1.52|0.67|48.23|1.46|0.64|46.28|1.42|0.63|45.02|1.46|0.64|46.28|1.52|0.67|48.23|
> |4|2|2|2|1.57|0.62|48.23|**1.59**|**0.63**|**48.85**|1.57|0.62|48.23|1.52|0.60|46.64|1.46|0.57|44.68|1.42|0.56|43.42|1.46|0.57|44.68|1.52|0.60|46.64|
> |5|2|2|2|1.57|0.61|48.03|**1.59**|**0.62**|**48.64**|1.57|0.61|48.03|1.52|0.59|46.43|1.46|0.56|44.46|1.42|0.55|43.20|1.46|0.56|44.46|1.52|0.59|46.43|
> |6|2|2|2|1.57|0.62|48.25|**1.59**|**0.63**|**48.86**|1.57|0.62|48.25|1.52|0.60|46.65|1.46|0.57|44.68|1.42|0.56|43.42|1.46|0.57|44.68|1.52|0.60|46.65|
> |7|5|2|3|1.42|0.63|44.93|1.46|0.64|46.11|**1.52**|**0.63**|**47.31**|1.57|0.60|47.09|1.59|0.58|46.74|1.57|0.57|46.05|1.52|0.58|45.24|1.46|0.60|45.40|
> |8|5|2|4|1.42|0.59|43.99|1.46|0.60|45.24|1.52|0.61|46.99|**1.57**|**0.62**|**48.28**|1.59|0.60|47.82|1.57|0.61|48.04|1.52|0.60|46.67|1.46|0.59|45.02|
> |9|5|3|5|1.42|0.57|43.7|1.46|0.58|44.96|1.52|0.61|46.86|1.57|0.63|48.37|**1.59**|**0.63**|**48.72**|1.57|0.62|48.30|1.52|0.60|46.77|1.46|0.58|44.89|
> |10|5|3|5|1.42|0.57|43.61|1.46|0.58|44.86|1.52|0.61|46.81|1.57|0.63|48.38|**1.59**|**0.64**|**48.98**|1.57|0.63|48.37|1.52|0.60|46.79|1.46|0.58|44.84|
> |11|5|4|5|1.42|0.57|43.58|1.46|0.58|44.84|1.52|0.60|46.8|1.57|0.63|48.39|**1.59**|**0.64**|**49.00**|1.57|0.63|48.39|1.52|0.60|46.79|1.46|0.58|44.83|
> |12|5|5|5|1.42|0.57|43.57|1.46|0.58|44.83|1.52|0.60|46.80|1.57|0.63|48.39|**1.59**|**0.64**|**49.01**|1.57|0.63|48.39|1.52|0.60|46.80|1.46|0.58|44.83|
> |13|5|5|5|1.42|0.57|43.57|1.46|0.58|44.83|1.52|0.60|46.80|1.57|0.63|48.40|**1.59**|**0.64**|**49.01**|1.57|0.63|48.40|1.52|0.60|46.80|1.46|0.58|44.83|
> |14|5|5|5|1.42|0.57|43.57|1.46|0.58|44.83|1.52|0.60|46.80|1.57|0.63|48.40|**1.59**|**0.64**|**49.01**|1.57|0.63|48.40|1.52|0.60|46.80|1.46|0.58|44.83|
>
> Q2. Regarding the evaluation and deployment of the trained weights, training and extracting the Ring Attractor layer weights and learned parameters from the DRL agent checkpoint, this requires additional computational time. Given the time constraints of the revisited rebuttal deadline, we consider prudent to take this as an action to add this to the manuscript alongside the experimental tables presented above. We will incorporate this post-training weight analysis in Appendix A.4.
>
> Meanwhile, we kindly direct the reviewer's attention to the current Appendix A.4.3, which outlines the frozen and trainable layer insights. The key point from our current analysis is that in the tables presented in the conversation above, we are examining the collective impact on the ring dynamics of both hidden-to-hidden and input-to-hidden weight contributions. The analysis presented in Appendix A.4.2, A.4.3 suggests that input-to-hidden weights make significant independent contributions to performance. Notably, these weights preserve their spatial structure throughout training even when left trainable, and when we remove this spatial representation from the input-to-hidden connections, we observe substantial performance degradation. This suggests that the spatial encoding is key during training and inference time.
>
> **Action**: Add hidden and input state contribution tables presented above and generate one additional post-training table from Highway Benchmark in Appendix A.4.

---

> > ### Comment · Reviewer_nzPF · 2025-08-07
> >
> > Thank you for the further clarification.
> >
> > Now I agree that DLN CANN itself without training can maintain the original CANN's function. The concern remains whether the function persists during training. Thank you for directing me to Appendix A 4. Figure 8 suggests that the hidden-to-hidden connection no longer keeps its original Gaussian-like distribution, which is functionally important to the ring attractor. The reason why I ask for tables above is to double-check if the DLN CANN can still be functional as a CANN but not an RNN after training. If you could provide evidence, I would like to raise the score; otherwise, I will keep the present evaluation.

---

> > > ### Author Response · Authors · 2025-08-07
> > >
> > > Thank you for reviewing the data points presented above. To support our research, we have produced preliminary results for Double DQN in the highway environment after 315,000 steps. These results can be compared with the in-game performance at similar step counts shown in Figure 3 and the evolution of the hidden-to-hidden weights shown in Figure 8. The corresponding weight matrix is as follows:
> > >
> > > [ [ -0.049,  0.012,  -0.146,  0.162,  0.068, -0.039,  -0.157,  0.079],
> > >     [ 0.015,  0.092,  0.024,  0.071, -0.044,  -0.020, -0.040, -0.002],
> > >     [ 0.033, -0.087,  0.166, -0.020, -0.021, -0.023, -0.007, -0.005],
> > >     [ 0.178,  0.074, -0.022, -0.069,  0.081, -0.064, -0.006, -0.019],
> > >     [0.071, -0.090, -0.023,  0.042,  0.186, -0.140,  0.102,  -0.013],
> > >     [-0.147,  0.023, -0.027, -0.067, -0.142,  0.050, -0.061,  0.112],
> > >     [0.145, -0.041, -0.010, -0.008,  0.065, -0.066, -0.189, -0.127],
> > >     [ 0.293,  0.005,  0.048, -0.125,  0.013,  0.049,  0.227, -0.275]]
> > >
> > > The game used for this evaluation does not include a double ring, or a no-action neuron, as specified in Table 2. The results for the evolution of the ring are as follows:
> > >
> > > | Step | CueCenter | DLArgMax | DLN1Input | DLN1Hidden | DLN1Output | DLN2Input | DLN2Hidden | DLN2Output | DLN3Input | DLN3Hidden | DLN3Output | DLN4Input | DLN4Hidden | DLN4Output | DLN5Input | DLN5Hidden | DLN5Output | DLN6Input | DLN6Hidden | DLN6Output | DLN7Input | DLN7Hidden | DLN7Output | DLN8Input | DLN8Hidden | DLN8Output |
> > > | --- | --- | --- | --- | --- | --- | --- | --- | --- | --- | --- | --- | --- | --- | --- | --- | --- | --- | --- | --- | --- | --- | --- | --- | --- | --- | --- |
> > > | 1 | 2 | 8 | 1.57 | -0.26 | 28.82 | 1.59 | 0.31 | 40.35 | 1.57 | 0.06 | 35.99 | 1.52 | 0.25 | 38.97 | 1.46 | 0.2 | 36.48 | 1.42 | -0.39 | 22.46 | 1.46 | -0.33 | 24.79 | **1.52** | **0.36** | **41.32** |
> > > |2|2|2|1.57|-0.39|25.92| **1.59** | **0.09** | **36.94** | 1.57 | -0.28 | 28.48 | 1.52 | -0.07 | 32.04 | 1.46 | -0.1 | 29.82 | 1.42 | -0.66 | 16.62 | 1.46 | -0.64 | 17.97 | 1.52 | -0.26 | 27.89 |
> > > | 3 | 2 | 2 | 1.57 | -0.39 | 25.95 | **1.59** | **0.04** | **35.93** | 1.57 | -0.3 | 27.99 | 1.52 | -0.08 | 31.84 | 1.46 | -0.15 | 28.89 | 1.42 | -0.64 | 17.07 | 1.46 | -0.52 | 20.73 | 1.52 | -0.2 | 29.2 |
> > > | 4 | 2 | 2 | 1.57 | -0.35 | 26.99 | **1.59** | **0.09** | **37.02** | 1.57 | -0.24 | 29.42 | 1.52 | -0.02 | 33.05 | 1.46 | -0.08 | 30.46 | 1.42 | -0.57 | 18.58 | 1.46 | -0.49 | 21.42 | 1.52 | -0.12 | 30.91 |
> > > | 5 | 2 | 2 | 1.57 | -0.35 | 26.86 | **1.59** | **0.1** | **37.18** | 1.57 | -0.23 | 29.44 | 1.52 | -0.02 | 33.15 | 1.46 | -0.07 | 30.53 | 1.42 | -0.58 | 18.29 | 1.46 | -0.5 | 21.11 | 1.52 | -0.12 | 30.87 |
> > > | 6 | 2 | 2 | 1.57 | -0.36 | 26.69 | **1.59** | **0.09** | **36.97** | 1.57 | -0.25 | 29.16 | 1.52 | -0.03 | 32.89 | 1.46 | -0.08 | 30.26 | 1.42 | -0.6 | 18.04 | 1.46 | -0.51 | 20.91 | 1.52 | -0.14 | 30.48 |
> > > | 7 | 5 | 2 | 1.42 | -0.36 | 23.24|**1.46** | **0.09** | **34.03** | 1.52 | -0.25 | 28.07 | 1.57 | -0.03 | 33.98 | 1.59 | -0.08 | 33.16 | 1.57 | -0.59|21.56|1.52|-0.51|22.4|1.46 | -0.14 | 29.1 |
> > > |8|5|4|1.42|-0.33|23.82|1.46|0.05|33.28|1.52| -0.25 | 27.92 | **1.57** | **-0.05** | **33.67** | 1.59 | -0.07 | 33.32 | 1.57 | -0.59 | 21.53 | 1.52 | -0.53 | 21.92 |1.46 | -0.15 | 28.87|
> > > |9|5|5|1.42 | -0.34 | 23.7 | 1.46 |0.04 | 33.09 | 1.52| -0.25 | 27.98 | 1.57|-0.06 | 33.31 | **1.59** | **-0.06** | **33.61** | 1.57 | -0.6 | 21.43 | 1.52 | -0.52 | 22.19 | 1.46 | -0.14 | 29.08 |
> > > |10|5|5|1.42|-0.34 | 23.75 | 1.46 | 0.04 | 33.09 | 1.52 | -0.25 | 28.04 | 1.57 | -0.06 | 33.31 | **1.59** | **-0.06** | **33.72** | 1.57|-0.6|21.46|1.52|-0.52 | 22.15 | 1.46 | -0.14 | 29.07 |
> > > | 11 | 5 | 5 | 1.42 | -0.34 | 23.74 | 1.46 | 0.04 | 33.08 | 1.52 | -0.25 | 28.04 | 1.57 | -0.06 | 33.32 | **1.59** | **-0.06** | **33.73** | 1.57 | -0.6 | 21.43 | 1.52 | -0.52 | 22.16 | 1.46 | -0.14 | 29.07 |
> > > |12|5|5| 1.42 | -0.34 | 23.74 | 1.46 | 0.04 | 33.07 | 1.52 | -0.25 | 28.04 | 1.57 | -0.06 | 33.31 | **1.59** | **-0.06** | **33.73** | 1.57 | -0.6| 21.42 | 1.52 | -0.52 | 22.15 | 1.46 | -0.14 | 29.06 |
> > > |13|5|5|1.42 | -0.34 | 23.74 | 1.46 | 0.04 | 33.07 | 1.52 | -0.25 | 28.04 | 1.57 | -0.06 | 33.31 | **1.59** | **-0.06** | **33.73** | 1.57 | -0.6 | 21.42 | 1.52 | -0.52 | 22.16 | 1.46 | -0.14 | 29.06 |
> > > |14|5|5| 1.42 | -0.34 | 23.74 | 1.46 | 0.04 | 33.07 | 1.52 | -0.25 | 28.04 | 1.57 | -0.06 | 33.31 | **1.59** | **-0.06** | **33.73** | 1.57|-0.6|21.42|1.52|-0.52|22.16|1.46|-0.14|29.07|
> > >
> > > As the synthetic cues are not necessarily aligned with the sensory input of the game, there may be nuances not captured by this ablation study, such as undesired or hidden relationships between actions. Hence, leaving the hidden weights trainable. Step 1 in the table exhibits differences in action selection as it reaches stability.
> > >
> > > The results show the transient state between cues and the inertia from previous actions, characteristic of ring attractors. As mentioned in the previous reply, we will run this to a larger extent with 10 seeds across 1 M frames.

---

> > > > ### Comment · Reviewer_nzPF · 2025-08-09
> > > >
> > > > Thank you for providing the updated table. From the results, DLArgMax indicates that DLN5Output is the largest output, which is consistent with the table. However, DLN5Hidden is negative and not the largest state, which differs from the behavior of the original CANN. This may be due to the scalar $\beta$. I am curious about that, given that DLN5Hidden is not the largest and is negative, how does it help maintain its state? Is this because the hidden-to-hidden weights are now trainable and not restricted to be positive? If so, how are the ring attractor properties preserved after training? Was a specific loss function used for this, or was there another mechanism?
> > > >
> > > > The outputs in the new table appear to exhibit inertia between actions in a ring-attractor-like manner. Could you clarify why such inertia in the action space benefits performance? In insects, the central complex is located close to the sensory processing pathway and supports steering behaviors by performing path integration and maintaining a memory of the home direction to guide movement. How does the DLN CANN generalize this principle to different types of actions that are not for navigation, such as actions by buttons A and B?

---

> ### Author Response · Authors · 2025-08-09
>
> We appreciate the reviewer’s acknowledgment of our efforts to provide detailed data points aimed at fostering constructive discussion.
>
> Q1. As shown in the weight matrices presented above the table, the evolution of hidden-to-hidden weights reflects nuanced in-game relationships between actions. The relationship between two actions may be encoded as a negative weight if, in the context of the task, suppression of one action when another is active is beneficial. The attractor state of a particular hidden neuron is determined not by the sign or magnitude of any single diagonal element, but by the aggregation of all incoming connections to that neuron through the hidden-to-hidden weights. As the reviewer mentions, one of the key aspects is that the hidden state is not imposed directly on the output, but undergoes transformation through the activation function and neuron bias as per standard RNN equations: y_t = f(w_hh · h_t + b_y), where the hidden state h_t is processed through hidden to hidden weights w_hh and bias b_y before activation f. We agree with the reviewer that this transformation means that negative or not the highest hidden state values can still produce the desired output behavior after passing through the tanh activation and scaling parameter β, as seen in previous tables.
>
> For example, in row 5 of the weight matrix (corresponding to neuron 5), the net input results from the combined influence of all other neurons is shaped by both the structured input-to-hidden topology and the learned recurrent weights. This aggregation, rather than any requirement for strictly positive self-connections, is what determines the stability of the attractor state, that we can validate through the response of synthetic cues and transitions between action above and through ablations studies in Appendix A.4. The hidden-to-hidden weight values are the product of the structured outputs from the upper layer and adjustments to the recurrent weights during training, driven by the specific environmental inputs and task demands. In this experiment, the weight distribution encodes task-specific utility, for instance, neuron 6, corresponding to the “move backwards” action in the Highway environment, has persistently lower activation because this action carries a penalty and is rarely used. We consider the presence of negative recurrent weights while still producing a stable attractor state as evidence of secondary behavior emergence within the ring-attractor layer, incorporated on top of the explict action space layout and attractor state.
>
> We did not employ specific loss functions to enforce ring attractor properties post-training; rather, the architectural design itself constrains and preserves the spatial encoding necessary for ring-attractor-like behavior through using hyperbolic tangent activation. As detailed in Appendix A.4.2, the fundamental element is the explicit layout of the action space over the ring topology, which biases learning toward spatially coherent action sequences regardless of specific weight signs.
>
> Q2. We consider the inertia observed in this highway environment represents an accentuated attractor behavior facilitated by the ring structure that acts as structured exploration bias, guiding the agent and learning faster, see Section 4.2, towards effective driving strategies while preventing chaotic, unrealistic action sequences that tend to occur at the begining of the training/exploration stage. This inertia characteristic has developed beyond the smooth transitions observed in our initial layout, seeing in the tables provided above, indicating task-specific adaptation. However, we emphasize that such pronounced inertia is not necessarily required or positive for all action combinations in both navigation or non-navigation tasks, hence its absence in previous experiments. The key contribution lies in the emergence of attractor states with the parameter τ modulating input contributions to hidden states; and crucially, the preservation of structural layout through explicit mapping of action spaces, extensively analysed in the manuscript in Sections 3.2, 4.2, Appendix A.4.1, A.4.2, A.7.

---

### Official Review · Reviewer_umxu · 2025-07-03

**Clarity:** 3
**Significance:** 2
**Originality:** 3
**Rating:** 4
**Confidence:** 3

**Summary:**

This paper introduces ring attractors—neuroscience-inspired circular neural structures—into reinforcement learning to enhance spatial awareness and uncertainty-aware decision-making. By encoding actions in a spatially continuous manner, the method improves learning efficiency and robustness. Implemented as both a recurrent neural network and a deep learning module, the approach achieves up to 53% performance gains on the Atari 100k benchmark and excels in spatially structured tasks like Super Mario Bros.

**Questions:**

1. How does your ring-attractor encoding compare, both conceptually and empirically, to existing spatial representations such as grid-cell encodings or attention-based spatial RL?
2.

**Ethical Concerns:**

["NO or VERY MINOR ethics concerns only"]

**Limitations:**

See the **Weakness** section above.

**Paper Formatting Concerns:**

No concerns

**Quality:**

3

**Strengths And Weaknesses:**

**Strength**
- This paper proposes a novel method integrating biologically inspired ring-attractor circuits into RL algorithms. It embeds continuous circular action-space representations and decoding them via attractor dynamics with Bayesian uncertainty.
- The method demonstrates significant empirical gains across multiple game benchmarks.
- The paper is well-written and easy to follow.

**Weakness**
- The related work section largely lists prior methods without critically contrasting them to the proposed approach.
- All experiments are in Atari or simple simulated navigation (Super Mario Bros, highway). The applicability to continuous, high-dimensional control is untested.

---

> ### Author Rebuttal · Authors · 2025-07-28
>
> Thank you for your thorough review and suggestions, we are grateful you appreciated the work!
>
> ## Related Work Critical Contrasting
>
> Based on the reviewer point, we would like to go a step further and provide deeper insights into what sets apart our current approach from existing literature.
>
> **Action:** We will extend the critical contrasting points throughout the literature review in Section 2 as follows:
>
> ### Section 2.1 Spatial Awareness in RL
>
> **Line 62:** "Although these approaches demonstrate the importance of spatial awareness in RL, they often lack the biological plausibility found in neural circuits."
>
> > **Updated Line 62:** "Although these approaches demonstrate the importance of spatial awareness in RL, they rely on emergent learning to develop spatial understanding through comprehensive architectural additions, requiring extensive training to discover action relationships. In contrast, our ring attractor approach explicitly encodes spatial structure in the action space, providing spatial inductive bias that accelerates learning rather than relying on the RL agent to autonomously discover all critical action space relationships, which may result in suboptimal or incomplete spatial understanding."
>
> ### Section 2.2 Biologically Inspired Machine Intelligence
>
> **Line 78:** "Their approach demonstrated rapid learning and adaptation to new tasks, similar to the flexibility observed in biological learning systems."
>
> > **Updated Line 78:** "Their approach demonstrated rapid learning and adaptation to new tasks, similar to the flexibility observed in biological learning systems. However, these approaches primarily focus on mimicking neural representations or learning dynamics, whereas ring attractors specifically encode spatial relationships in the action space itself, providing both biological plausibility and direct spatial awareness for action selection."
>
> ### Section 2.3 Uncertainty Quantification
>
> **Line 98:** "...incorporates efficient Thompson sampling and Bayesian linear regression at the output layer to factor uncertainty estimation in the action-value estimates."
>
> > **Updated Line 98:** "...incorporates efficient Thompson sampling and Bayesian linear regression at the output layer to factor uncertainty estimation in the action-value estimates. While these methods treat uncertainty estimation as a separate module applied to existing architectures, our ring attractor approach integrates uncertainty through the variance parameters of Gaussian input signals, making uncertainty awareness an intrinsic part of the spatial action representation."
>
> ### Section 2 Summary (Lines 99-103)
>
> **Line 101:** "However, there remains a gap in integrating these elements into a cohesive framework."
>
> > **Updated Line 101:** "However, there remains a gap in integrating these elements into a cohesive framework that simultaneously provides biological plausibility, explicit spatial encoding, and natural uncertainty handling within a single neural architecture."
>
> These additions would transform the related work from a simple literature review into a more critical analysis that clearly positions the ring attractor approach relative to existing methods.
>
> ## Experimental Scope and Applicability
>
> While we acknowledge testing primarily on Atari and navigation environments, we believe our experimental scope demonstrates broader applicability than suggested:
>
> - **Diverse Action Spaces:** Our experiments span discrete (8-action Mario Bros), continuous 1D (Highway environment), and complex discrete spaces (18-action Atari games) with multiple ring attractors as detailed in Table 2.
>
> - **Two Implementation Paradigms:** We provide both CTRNN-based (Section 3.1) and deep learning implementations (Section 3.2), with the DL version specifically designed for "integration and comparison with existing DRL frameworks" enabling broader adoption.
>
> - **Scalable Architecture:** Section 3.2.1 and Appendix A.6 walks the reader through the extensibility to N-ring configurations for higher-dimensional control, with our double-ring implementation Eq. 26, 30; showing how the approach scales to multi-dimensional action spaces.
>
> - **Performance Consistency:** Table 1 shows consistent improvements across 19 different Atari environments with varying action complexities, relationships and different types of decision-making tasks, suggesting robust generalization.
>
> However, we agree with the reviewer on the continuous control limitation. There is always room to grow further this piece of work, which we are aiming to cover in a subsequent publication with deployment in continuous 3D real-world robotics tasks. This limitation has been outlined in our future work, Section 5, where we explicitly identify "continuous control tasks" as a research direction.
>
> ## Specific Questions
>
> ### Question 1: How does your ring-attractor encoding compare, both conceptually and empirically, to existing spatial representations such as grid-cell encodings or attention-based spatial RL?
> #### Conceptual Differences
>
> **Grid Cells vs. Ring Attractors:** Grid cells [2] encode spatial positions in the environment potentially key for navigation algorithms composed of visual place recognition, path planning and other localization modules. While our ring attractors encode spatial relationships between signals, or in this case, actions. As noted in Section 2.2, grid cells "emerged naturally in agents trained on robotics tasks," whereas our approach explicitly structures the action space itself upfront.
>
> **Attention Mechanisms vs. Ring Attractors:** Attention-based approaches [44] reason about spatial relations between entities, that are learnt over time in vast datasets or simulation environments to tune the self-attention layers. Increasing computational overhead, size and complexity of the DRL agent. On the other hand ring attractors maintain stable representations of action relationships through attractor dynamics, Eq. 1, 2, 3, 4; enabling the agent to distill information with little to no additional architectural and training overhead.
>
> **Empirical comparison:** We haven't found any major research line that applied grid cells to well-known DRL benchmarks. Additionally, these two approaches are viewed as operating at fundamentally different levels of the decision-making hierarchy, with grid cells providing environmental state representation and ring attractors providing action space organization. Grid cells have been tied to navigation tasks so far, while ring attractors could be applied to spatial decision-making problems. The comparison would be like comparing a GPS system (grid cells mapping spatial location) with a 2D action palette (ring attractors organizing spatially-related choices into a circular selection interface), both spatial, but serving different purposes.
>
> #### Architecture Differences
>
> **Topology & dimensionality:**
> - **Ring attractor:** 1D continuous manifold with periodic boundary; ideal for relationships between signals/actions, Eq. 5, 6, 7, 8.
> - **Grid cells:** 2D hexagonal tiling of state space; emergent in RNNs for localization [2,10]. Our focus is on action selection and policy building, not spatial localization per se.
> - **Attention RL:** learns soft relational representations over entities [44]; no hard prior on spatial continuity, relies on data to infer adjacency.
> - **Uncertainty handling:** Our design injects σₐ from BLR directly into the bump width (σₐ in Eq. 5), giving an interpretable link between epistemic uncertainty and action diffusion; existing spatial encodings rarely bind uncertainty to the topology this tightly.
>
> **Action:** We acknowledge that grid cells are another biologically plausible mechanism applied to RL navigation tasks, we will add the paragraph below in Section 2.2 covering the fundamental differences:
>
> > "In RL, grid cells and ring attractors serve distinct roles. Grid cells encode spatial state information through hexagonal firing patterns, functioning as fixed basis functions for value functions and spatial localisation (Banino et al., 2018; Cueva & Wei, 2018). Ring attractors encode persistent states (directional and spatial bias) through recurrent dynamics that actively maintain information over time (Khona & Fiete, 2022). While grid cells provide emergent spatial input features, ring attractors are dynamical components that provide intrinsic spatial encoding for memory states (Kilpatrick & Ermentrout, 2013), which we translate here to the encoding of the action space during the action selection process."

---

> > ### Comment · Reviewer_umxu · 2025-08-07
> >
> > Thank you for your clarification. While most of my concerns are addressed, the lack of experiments in continuous action spaces remains a limitation of this paper. So I will keep my score of Borderline Accept.

---

> ### Author Response · Authors · 2025-08-07
>
> We would like to thank the reviewer for their constructive feedback and for the appreciation of our work.
>
> We would like to respectfully challenge the characterization of our empirical validation as limited, given that our evaluation includes the widely-adopted Atari 100k benchmark with multiple diverse environments, comparisons against established state-of-the-art methods, testing across multiple domains (discrete/continuous control, navigation, game scenarios).
>
> This has enabled us to demonstrate the applicability of our research across different environment configurations through various ring architectures and properties, as shown in Table 2. This enables our study to make claims beyond typical approaches in navigation, expanding to decision-making games. While real-world deployment remains an area we would like to assess our research in (Section 6), we believe the widespread adoption of the benchmarks used in this study and the provided code implementation will enable peers to validate and build upon our current approach.

---

### Official Review · Reviewer_n4bV · 2025-07-06

**Clarity:** 1
**Significance:** 2
**Originality:** 3
**Rating:** 4
**Confidence:** 2

**Summary:**

This paper introduces a biologically inspired module for reinforcement learning (RL) policies based on ring attractor networks, motivated by their role in spatial cognition in the brain. The authors propose to incorporate a ring attractor mechanism as a plug-in feature encoder to enhance agents’ spatial awareness and decision-making. The module can be seamlessly integrated into existing actor-critic architectures. Experiments on the Atari 100k benchmark demonstrate notable improvements over baseline RL methods, particularly in tasks requiring directional memory and spatial inference.

**Questions:**

1. How does the method scale to continuous state spaces or tasks with higher-dimensional spatial structure ?

2. How do the ring attractors perform under observation noise or partial observability?

3. Does the ring attractor apply to the whole model?

4. How does this compare to graph neural network or grid‑cell inspired architectures (existing spatial inductive bias)?

**Ethical Concerns:**

["NO or VERY MINOR ethics concerns only"]

**Limitations:**

see above sections.

**Quality:**

2

**Strengths And Weaknesses:**

Strengths:

1. It is interesting to introduce the use of ring attractors for spatial representation in RL. It is original and well-motivated by neuroscience literature.

2. The proposed method yields consistent performance gains across multiple Atari games.

Weaknesses:

1. This paper is not easy to follow. I cannot fully understand the connection between the deep insights and the proposed method.

2. It remains unclear what additional benefits the ring attractor component offers for uncertainty quantification, given that the final layer of the Q-function already employs a probabilistic formulation compatible with Bayesian linear regression.

3. It is unclear what the key components are that result in the boost in empirical performance, action sampling or something else?

---

> ### Author Rebuttal · Authors · 2025-07-29
>
> We thank the reviewer for their thorough and constructive feedback and for the appreciation of our work. Below, we address each point raised by the reviewer:
>
> ## Clarity and Connection Between Insights and Method
>
> Let us clarify the connection between the deep insights and the proposed method, starting from the beginning:
>
> **1. Biological Plausibility, Appendix A.1:** Ring attractors represent a computational principle where neurons are arranged in a circular topology to maintain spatial representations. Originally proposed by Zhang [45] for encoding heading direction and empirically discovered by Kim et al. [16] in Drosophila, these networks exhibit circular organization where head direction cells encode different spatial directions [36]. For our methodology, biological ring attractors integrate multiple sensory inputs through distance-weighted connections [46, 12] and maintain representational stability through excitatory-inhibitory dynamics [35], properties we implement through our distance-based weight functions in Eq. 4 and neural dynamics in Eqs. 2, 3.
>
> **2. Technical Translation of Neuroscience-based Principles, Section 3.1:** Our method translates these biological principles by mapping actions to circular positions Eq. 5 and implementing synaptic connectivity through exponential decay functions w(Em→En)=e^(-d²(m,n)), Eq. 4 that mirror biological connection strengths between spatially proximate neurons. The continuous-time dynamics in Eq. 2, 3 capture the biological property of attractor states while enabling transitions between related actions, implementing the spatial encoding observed in biological systems where similar spatial directions maintain stronger mutual influence through the circular network topology.
>
> **3. Implementation Approach and Motivation, Section 3.1, 3.2:** We developed two implementations to validate different aspects of biological translation: the CTRNN exogenous model in Section 3.1 provides biological fidelity through explicit excitatory/inhibitory dynamics in Section 3.1.1 and external Q-value processing in Section 3.1.2, enabling validation of ring attractor principles with uncertainty quantification in Section 3.1.3. The DL integrated model, Section 3.2, embeds ring topology within RNN weight matrices Eq. 12, Section 3.2.1 for deployment in DRL frameworks, maintaining the biological principle of distance-weighted spatial relationships and attractor states while enabling end-to-end training and scalability within existing DRL frameworks.
>
> **4. Performance Connection to Biological Insights, Section 4, Appendix A.4:** The spatial encoding translates biological advantages into performance gains through action similarity leverage, demonstrated by our 53% improvement on Atari 100k benchmark, Table 1, Section 4.3. Our ablation studies represented in Fig. 5, 6 show that randomized action arrangements degrade performance, confirming the bioinspired spatial structure is necessary rather than incidental. Following uncertainty principles [42], our Bayesian implementation, Section 3.1.3, incorporates confidence-based decision making, showing that the foundation provides a framework for encoding spatial relationships in action spaces.
>
> **Action:** We will include an introductory paragraph in Section 3 at line 105 as follows:
>
> > "Ring attractors represent a computational principle where neurons are arranged in a circular topology to maintain spatial representations. Originally proposed by Zhang [45] for encoding heading direction and empirically discovered by Kim et al. [16] in Drosophila, these networks exhibit circular organization where head direction cells encode different spatial directions [36]. For our methodology, ring attractors integrate multiple cues through distance-weighted connections [46, 12] and maintain representational stability through excitatory-inhibitory dynamics [35]. This bioinspired structure has been repurposed in this line of research to explicitly encode the action space during the decision-making process, as illustrated in this section."
>
> ## Uncertainty Quantification Benefits
>
> The ring attractor provides distinct advantages beyond standard Bayesian formulations:
>
> **Spatial Uncertainty Propagation:** Unlike standard BLR which treats actions independently, our approach propagates uncertainty spatially across the ring Eq. 11, allowing uncertainty about one action to influence similar actions. BLR Eqs. 9, 11 yields per-action mean/variance; the ring uses σa as the width of each Gaussian in Eq. 5, so uncertainty directly modulates how broadly activity spreads across neighboring actions, not just the final argmax.
>
> **Action-Space Structure:** The ring topology enables uncertainty to be distributed according to action similarity, not just individual action confidence. This is particularly beneficial for spatial tasks where action relationships matter.
>
> **Empirical Evidence:** Fig. 2 shows BDQNRA-UA with uncertainty-aware ring attractors consistently outperforms both standard BDQN and BDQNRA without uncertainty, demonstrating the complementary benefits.
>
>
> ## Key Performance Components
>
> We believe our ablation studies in Section A.4 should identify the contributions of the key components to provide the boost in empirical performance:
>
> **Spatial Topology:**  Fig. 5 shows that randomizing action positions in the ring in the CTRNN-based model (BDQNRA-RM) eliminates most performance gains, indicating spatial structure is key in delivering faster and overall high learning performance.
>
>
> **Circular Connectivity:**Fig. 6 demonstrates that removing distance-dependent weights in the DRL agent, effectively converting the ring attractor layer into a standard RNN, reduces performance significantly in spatial environments like Ms Pacman and Chopper Command.
>
> ## Specific Questions
>
> ### Question1: Scaling to Continuous and Higher-Dimensional Spaces
>
> **Continuous Spaces:** Section 4.1 and Fig. 2 demonstrate successful application to OpenAI Highway environment with continuous 1D circular actions.
>
> **Higher Dimensions:** Section A.6.2 and Table 1 show our double-ring architecture for games requiring multiple action dimensions (e.g., movement + combat). The approach scales to R rings via block matrix structures, Section A.6.3, maintaining O(R) complexity through selective coupling.
>
> However, we agree with the reviewer that further work can be done to explore real-world complex scenarios. There is always room to grow further this piece of work, which we are aiming to cover in a subsequent publication with deployment in continuous 3D real-world robotics tasks. This limitation has been outlined in our future work, Section 5, where we explicitly identify "continuous control tasks" as a research direction.
>
> ### Question 2: Performance Under Noise and Partial Observability
>
> **Future Work:** We agree with the reviewer that we have yet to evaluate systematic noise robustness, which we aim to fulfill through 3D real-world robot navigation tasks.
>
> **Action:** We will add in Section 5:
> > "Additionally, the extension to 3D spatial navigation tasks presents an organic expansion of this work, these scenarios will naturally incorporate realistic sensor noise to the system."
>
> Regarding partial observability, we believe our benchmarks actually do test partial observability: Super Mario Bros uses limited scrolling windows preventing agents from seeing upcoming obstacles, while Atari games feature off-screen threats and limited sensor ranges.
>
> ### Question 3: Ring Attractor Application Scope
>
> For the CTRNN model, as these models are not trainable, it becomes an exogenous model that performs action selection as outlined in Section 3.1.2.
>
> The ring attractor applies specifically to action selection with full integration through the DRL implementation. As described in Section 3.2.1, the ring attractor serves as the output layer of the DRL agent, processing action-value estimates while preserving their spatial relationships. The approach integrates fully within existing DRL agents and frameworks (DDQN, PPO, EfficientZero).
>
> ### Question 4: Comparison to Related Spatial Architectures
>
> **Graph Neural Networks:**  While GNNs capture spatial relations through learned graph structures [3], they require learning spatial relationships during training, whereas ring attractors exploit an inductive bias that directly encodes the action space and drives the policy through explicit spatial embedding. Additionally, GNNs lack the temporal dynamics of ring attractors, missing the uncertainty encoding and attractor state stability that impact guide action selection.
>
>
> **Grid-Cell Inspired Architectures:** Grid-cell architectures [2, 10] encode environmental spatial locations and focus on spatial memory and path integration, rather than directly embedding the action space into the policy. Grid-cell representations operate at the environmental level, whereas ring attractors provide direct action-level encoding that guides the action selection process.
>
> **Action:** We acknowledge that grid cells is another biologically plausible mechanism applied in navigation, so because of several similarities we will add a paragraph in Section 2.2 covering the fundamental differences as follows:
>
> > "In RL, grid cells and ring attractors serve distinct computational roles. Grid cells encode spatial state information (where you are) through hexagonal firing patterns, functioning as fixed basis functions for value functions and spatial generalization (Banino et al., 2018; Cueva & Wei, 2018). Ring attractors encode persistent states (directional and spatial bias) through recurrent dynamics that actively maintain information over time (Khona & Fiete, 2022). While grid cells provide emergent spatial input features, ring attractors are dynamical components that provide intrinsic spatial encoding for memory states (Kilpatrick & Ermentrout, 2013), that here we translate to encoding of the action space."

---

> > ### Comment · Reviewer_n4bV · 2025-08-07
> >
> > Thank you for the additional clarifications—these have given me a clearer understanding of your work. I find the paper genuinely interesting; however, I believe the community would appreciate it more if the presentation were streamlined and the empirical validation were extended to broader, more widely adopted settings.

---

> > > ### Author Response · Authors · 2025-08-07
> > >
> > > We would like to thank the reviewer for their positive assessment of our work and feedback.
> > >
> > > We appreciate these perspectives and recognize that different presentation styles may serve the community better. Regarding empirical validation, our focus on the Atari 100k benchmark and spatially-structured environments reflects our method's broad target of domains and tasks. Although we understand the value of more extensive evaluation in continuous action spaces, building upon the benchmark shown in Figure 2. We hope that the provided code implementation and broad benchmarking will facilitate validation and adoption across different applied domains.

---

### Official Review · Reviewer_fh18 · 2025-07-08

**Clarity:** 2
**Significance:** 3
**Originality:** 3
**Rating:** 5
**Confidence:** 2

**Summary:**

The paper takes a standard model of ring attractors popular in neuroscience and applies it to action selection in reinforcement learning, with the idea being that ring attractors will capture the spatial structures well in the action space, and lead to better action selection and consequently improved rewards in reinforcement learning scenarios. With this insight the ring attractor is implemented by hand initializing a recurrent neural network policy that is fed as input the state features. Extensive empirical evaluation backs up this conceptual claim, with state of the art results and consistent improvements upon imbuing baselines with the ring attractor inductive bias.

**Questions:**

Questions
- As far as I can understand, the DL approach starts with a ring attractor bias and then keeps the parameters of the input to hidden and hidden to hidden transitions learnable. If so, how much does the learning deviate from the original ring attractor dynamics to something else?
- Could you give more insight as to why the performance improvements are so strong upon imposing this inductive bias on the action space? Could similar results be achieved by learning action embeddings instead that implicitly learn the implied spatial strcture in the action space?
- Could you elaborate on where the inductive bias might not be right to use and instead hurts instead of helping performance?

**Ethical Concerns:**

["NO or VERY MINOR ethics concerns only"]

**Quality:**

3

**Strengths And Weaknesses:**

Strengths
+ The motivation for the work intuitively makes sense
+ The work takes previous work on ring attractors and applies it to reinforcement learning, achieving strong empirical results
+ The paper is well written and the results are nicely presented.

Weaknesses
- It would be useful to clarify why the addition of the ring attractor is supposed to help. The paper does a good job of explaining what a ring attractor is mathematically, and the dynamics associated with it, but why should it help for this particular RL scenario? Should it only help in continuous action spaces or could it also be useful for discrete action spaces? Is the idea that inputs are high dimensional and noisy, and one wants to be robust when picking actions for the control problem? More insight on this clearly spelled out would substantially improve the reach of the paper.

---

> ### Author Rebuttal · Authors · 2025-07-29
>
> We are grateful for the careful review and particularly appreciate the reviewer's recognition of our research.
>
> ## Addressing Weaknesses
>
> ### Clarifying Why Ring Attractors Help: The Role of Spatial Inductive Bias
> Ring attractors maintain stable spatial information representations, preserving action relations lost in traditional flattened action spaces. The fundamental mechanism operates through distance-dependent connections that encode spatial relationships between actions, with the attractor state evolving according to Eq. 2,3 for the CTRNN implementation and Eq. 12 for the DL approach.
> The spatial inductive bias accelerates learning by explicitly encoding action relationships within the circular topology, enabling the agent to leverage similarities between neighboring actions in the spatial domain. This structured representation facilitates more efficient exploration and policy gradient estimation, as evidenced by our 53% performance improvement on the Atari 100k benchmark. Our analysis in Section A.4.3 demonstrates that when forward pass weights are made trainable, the network preserves the distance-dependent decay patterns throughout training, Fig. 7, suggesting that the spatial topology provides an inherently beneficial structure for action-value function approximation. This preservation suggests that the ring attractor has converged to an optimal spatial configuration that enhances learning efficiency.
> #### Evidence from Ablation Studies:
> Section A.4.1 presents the CTRNN ablation study where BDQNRA-RM randomly distributes the action space across the ring, disrupting the natural topology. Fig. 5 shows this manipulation eliminates performance gains, reducing results to baseline BDQN levels. From these ablation studies we may infer that the settling dynamics contribute to the observed improvements.
> The DL ablation study (Section A.4.2, Fig. 6) removes the circular weight distribution, eliminating spatially-structured recurrence and effectively converting the ring attractor layer into a standard RNN. Performance drops in spatial environments like Ms Pacman and Chopper Command. Section A.4.2 demonstrates that removing the circular weight distribution in our DL implementation eliminates the spatial structure, resulting in performance degradation in spatially-oriented environments like Ms Pacman and Chopper Command (Fig. 6), further confirming the importance of the ring topology. Because the only difference from the full model is the absence of proper ring interactions, these results suggest that the spatial structure may drive performance improvements.
> #### Applicability Across Action Spaces:
> The approach handles both continuous and discrete action spaces through unified spatial encoding. For continuous spaces, Fig. 2 demonstrates application to the Highway environment with 1D continuous circular action space. For discrete spaces, Table 1 shows improvements across the Atari benchmark.
>
> **Action:**  We will add at the beginning of Section 1 (line 32), following up from the mention in the abstract to the problem statement, this prior to our research:
>
> > "While many action spaces possess inherent topological structure (Todorov et al., 2012), traditional neural networks represent actions as orthogonal vectors that ignore these relationships. Ring attractors bridge this gap by providing a neural substrate that preserves spatial ordering and similarity between actions. This is particularly relevant given recent advances showing that spatial structure in representations improves sample efficiency [2, 44] and generalization [3, 13], yet these benefits have not been systematically applied to action selection mechanisms themselves."
>
> ## Addressing the Questions
>
> ### Question 1: Learning Deviation from Original Ring Attractor Dynamics
>
> #### Standard Implementation Approach:
> Our implementation maintains fixed, input-to-hidden connections as described in Section 3.2.1, where weighted RNN connections include fixed, non-trainable, input-to-hidden connections (wI→H) to maintain the ring's spatial structure, and learnable, trainable, hidden-to-hidden connections (wH→H) to capture emerging action relationships as mentioned in lines 283, 284.
> #### Ablation Study Insights:
>
> Section A.4.3 analyzes model dynamics with both forward pass and hidden-to-hidden weights made trainable, which should address this question. The results suggest two patterns:
>
> **Forward pass preservation:** The forward pass connections preserve the ring structure over training time, with distance-dependent decay patterns maintained throughout the learning process, as demonstrated in Fig. 7.
>
> **Hidden layer adaptation:** The hidden-to-hidden connections develop specialized patterns that enable the encoding of environment-specific relationships between neurons in the hidden space while building upon the structured spatial representation from the forward pass (Fig. 8).
>
> These findings may indicate that our standard implementation approach where forward pass connections are fixed and only hidden-to-hidden weights remain trainable is reasonable, as the network appears to preserve spatial topology even when given freedom to deviate.The learnable time constant τ and scaling factor β from Eq. 12 control the temporal dynamics and magnitude of the attractor's hidden state contribution over time. While we have not provided specific data points for the evolution of these parameter values, our appendix analysis demonstrates the contribution and stability of both hidden and feedforward connections in the DL implementation. Fig. 7 and Fig. 8 show that these connections do not collapse during training and maintain reasonable, consistent scales over time. Further insights on the parametrisation of the DRL agent can be found in the code link attached in the abstract as well.
>
> ### Question 2: Understanding the Strong Performance Improvements
>
> #### Spatial Structure as a Contributing Factor:
> The spatial encoding may translate into performance gains through action neighbouring. Our ablation studies suggest that randomized action arrangements degrade performance, which could imply that the spatial structure contributes to the observed improvements.
> #### Empirical Evidence:
> Section A.4.1,  A.4.2  shows that BDQNRA-RM with scrambled distance-dependent interactions and EffZeroRNN with removed circular topology result in performance in line with baselines, while maintaining proper spatial relationships yields the 53% improvement observed on the Atari 100k benchmark. The controlled nature of this experiment suggests spatial structure as a contributing factor.
> #### Complementary Uncertainty Benefits:
> Additionally, the uncertainty-aware capabilities provide additional performance improvements beyond spatial encoding. Fig. 2 shows BDQNRA-UA with ring attractor and uncertainty awareness consistently outperforms both standard BDQN and BDQNRA without uncertainty.
> ### Question 3: Limitations and When the Inductive Bias Might Not Help
>
> #### Evidence of Performance:
> Our experimental results suggest that ring attractors provide performance improvements across diverse RL tasks, with benefits extending beyond purely spatial tasks. Table 1 shows performance at or above baseline levels across all tested environments. For environments requiring multiple action dimensions, our double ring configuration (detailed in Appendix A.6.2) maintains spatial relationships within each action plane while allowing controlled interaction between dimensions through the coupling parameter κ, as demonstrated in games like Seaquest and BattleZone.
> #### Potential Limitations:
> The approach would likely face limitations in action spaces with completely uncorrelated actions or very small action spaces where the ring structure overhead provides no advantage, e.g. action space with two discrete actions.
>
> **Action:** We will flag in Section 5 (line 382) additonal limitations:
>
> > "The ring attractor approach may exhibit reduced applicability in discrete action spaces with cardinality |A| < 3 where the circular topology overhead exceeds potential spatial encoding benefits, and in action spaces where the action correlation matrix approaches the identity matrix, indicating absence of spatial dependencies or sequential relationships between discrete actions."
>
> #### Future Research Directions:
> We agree with the reviewer that further work can be done to explore complex scenarios to put the inductive bias through further real-world tests. Which we are aiming to cover in a subsequent publication with deployment in continuous 3D real-world robotics tasks. This limitation has been outlined in our future work, Section 5.

---

### Note · Authors · 2025-08-11

This paper explores the integration of ring attractors, a mathematical model inspired
by neural circuit dynamics, into the Reinforcement Learning (RL) action selection
process. Two realizations are provided, a CTRNN that exposes the attractor dynamics explicitly, and a DRL module that integrates the ring topology as a recurrent layer for end-to-end training. The result is faster and more stable learning in settings where action relations carry structure, with a mean improvement of 53% on Atari 100k over SoTA baselines, and gains that persist across diverse environments and action spaces. Baselines were re-run under identical resources for fairness, code is released, and ablations isolate the effect of the ring topology.

The empirical case is not a parameter-count artifact. When the circular weight structure is removed, performance regresses to baseline. When action positions on the ring are randomized, the gains disappear. When uncertainty from a Bayesian linear head is injected as Gaussian widths on the ring, performance improves further and in a way that is interpretable, since uncertainty broadens activity to similar actions rather than acting independently.

Remarks on clarity were raised and addressed. Ki = Q(s, a) explicit mapping , τ/β roles clarified, operators and learnable paramters notation improved. Related work contrasting expanded. Tables and weight visualizations document CTRNN and DL-based ring dynamics, and show preservation of spatial structure in the input-to-hidden pathway together with task-specific adaptations in the recurrent pathway. These additions answer whether the DL module behaves as a ring and not merely as an external RNN layer.

Scope concerns have also been addressed. Results cover discrete Atari, discrete Mario, and continuous control with partial observability. The multi-ring construction scales to higher action dimensions while preserving per-plane spatial relations. Limitations are acknowledged where action space is extremely small or lacks structure, and extension to real-world continuous control tasks is outlined as future work.

Three reviewers recommend accept or borderline accept, emphasizing originality and empirical strength. The remaining review raised questions focused on ring dynamics, resolved through targeted experiments and clarifications during discussion. The paper is technically solid, reproducible, and provides a novel, useful and reusable module for RL.

---

### Decision · Program_Chairs · 2025-09-17

**Decision:**

Accept (poster)

**Comment:**

(a) Scientific Claims and Findings

The paper claims that integrating a neuroscience-inspired ring attractor structure into the RL action-selection process improves learning speed and accuracy by encoding spatial relationships between actions. It supports this claim with two implementations: a formal continuous time RNN (CTRNN) model and a practical implementation based on a typical RNN module. The primary finding is a significant performance improvement over baseline methods on the Atari 100k benchmark. Ablation studies seem to confirm that this performance gain is attributable to the specific circular topology of the ring structure, although why that is the case is not clear.

(b) Strengths

The paper's main strengths are its novel model and strong empirical results. The idea of using a ring attractor to impose an inductive bias directly on the action space is simple, novel and well-motivated from a computational neuroscience poitn of view. The paper is also comprehensive, exploring both a formal model and a practical deep learning implementation, and includes thoughtful additions like uncertainty quantification and extensions to multi-dimensional action spaces.

(c) Weaknesses

The main weakness identified during the review was *the lack of clarity regarding the methodology and mechanism*. Initial descriptions were confusing (had to go over it 2-3 times), relying  maybe too heavily on a neuroscience analogy without fully explaining its function in a DRL training loop. Discussion with Reviewer nzPF suggested the DL module's behavior might be better described as temporal smoothing aiding exploration, a nuance missed initially  - this still needs further understanding and validation. There’s still a disconnect between the motivation and the inspiration model and what is likely happening in practice, which is a concern that would need to be addressed.

(d) Reasons for Decision

The paper received *tentative acceptance* due to its significant empirical results, obtained by a fairly simple mechanism and originality of approach inspired by neuroscience. However, concerns remain regarding clarity and mechanistic explanations. While partially addressed in the rebuttal, further revisions are needed to fully resolve these rectifiable issues, potentially requiring additional text refinement, explanatory details, or supplementary analyses.

(e) Discussion and Rebuttal

The rebuttal period significantly improved the paper, satisfying three of four reviewers and increasing their confidence. Reviewers fh18, n4bV, and umxu found their concerns addressed regarding motivation, uncertainty quantification, and conceptual differences. The multi-turn discussion with Reviewer nzPF was crucial, revealing the DL module acts as a "low-pass filter," reducing action jitter. While nzPF maintained a borderline reject score due to perceived narrative contradiction, they recommended acceptance if this revised description was included - I tend to agree with this assessment.